# Non-Parametric Probabilistic Robustness:
# A Conservative Risk Estimator under Unknown Perturbation Distributions

**Zheng Wang** [1]   **Yi Zhang** [1]   **Siddartha Khastgir** [1]   **Carsten Maple** [1]   **Xingyu Zhao** [1][2]

## Abstract

Deep learning (DL) models, despite their remarkable success, remain vulnerable to small input perturbations that can cause erroneous outputs, motivating the recent proposal of probabilistic robustness (PR) as a complementary alternative to adversarial robustness (AR). However, existing PR formulations assume a fixed and known perturbation distribution, an unrealistic expectation in practice. To address this limitation, we propose non-parametric probabilistic robustness (NPPR), a more practical PR metric that does not rely on any predefined perturbation distribution. Following the non-parametric paradigm in statistical modeling, NPPR learns an optimized perturbation distribution directly from data, enabling conservative PR evaluation under distributional uncertainty. We further develop an NPPR estimator based on a Gaussian Mixture Model (GMM), covering various input-dependent and input-independent perturbation scenarios. Theoretical analyses establish the relationships among AR, PR, and NPPR. Extensive experiments on CIFAR-10, CIFAR-100, and Tiny ImageNet across ResNet18/50, WideResNet50 and VGG16 validate NPPR as a more practical robustness metric, showing conservative (lower) PR estimates compared to assuming those common perturbation distributions used in state-of-the-arts.

## 1. Introduction

Deep learning (DL) models, despite their success across perception, language, and control tasks, are known to be vulnerable to small input perturbations that can drastically alter their outputs. Such perturbations, commonly referred to as adversarial examples (AEs), reveal a fundamental weakness in modern DL models. Since the first discovery of the phenomenon of AEs (Szegedy et al., 2014; Goodfellow et al., 2015; Papernot et al., 2016), *robustness* has emerged as one of the most actively studied properties. Among them, adversarial robustness (AR) (Huang et al., 2020; Meng et al., 2022; Zühlke & Kudenko, 2024; Chakraborty et al., 2021) is arguably the most extensively studied, focusing on a model's ability to withstand *deterministic* and *worst-case*[1] AEs deliberately crafted by malicious attackers from a security perspective. In contrast, probabilistic robustness (PR) (Webb et al., 2019; Weng et al., 2019; Wang et al., 2021a; Pautov et al., 2022; Couellan, 2021; Baluta et al., 2021; Tit et al., 2021; Zhang et al., 2023; Tit et al., 2023; Zhang et al., 2024b; Robey et al., 2022; Zhao, 2026; Zhang et al., 2025; 2026) has recently been proposed as a complementary notion, not only from the reliability perspective (concerning average risk (Webb et al., 2019; Wang et al., 2021a; Zhao, 2026)) but also as an extension of the security viewpoint, aiming to quantify the *likelihood* that a DL model maintains correct behavior under *random* perturbations. Such random perturbations may arise from natural stochastic sources, e.g., sensor white noise in benign operational environments, or from unsophisticated attackers who rely on brute-force random-noise strategies rather than carefully optimized manipulations (i.e., attacks typically proposed in AR studies).

Intuitively, PR addresses the question: "What is the likelihood of encountering AEs under stochastic perturbations?". Such stochastic perturbations should be generated from *a given distribution*, referred to as the "input model" in, e.g., (Webb et al., 2019; Weng et al., 2019; Zhang et al., 2024b). Indeed, this perturbation distribution is assumed to be known and fixed in the very first formal definition of PR (Webb et al., 2019) and has since been adopted by *all* subsequent studies in both PR assessment (Webb et al., 2019; Weng et al., 2019; Wang et al., 2021a; Pautov et al.,

---

[1]The terms "worst-case" and "deterministic" indicate that AR studies typically aim to find the optimized AE, e.g., the one that maximizes loss or lies closest to the original input within a norm-ball. Note, although some approaches used to detect such deterministic/worst-case AEs involve *stochastic* procedures (e.g., random starts in PGD attacks), these are normally algorithmic choices for numerical optimisation, which does not change the fact that the underlying optimal AE is deterministic.

[1]WMG, University of Warwick, Coventry, United Kingdom [2]Wuhan University, Wuhan, China. Correspondence to: Xingyu Zhao <xingyu.zhao@warwick.ac.uk>.

*Proceedings of the 43rd International Conference on Machine Learning*, Seoul, South Korea. PMLR 306, 2026. Copyright 2026 by the author(s).

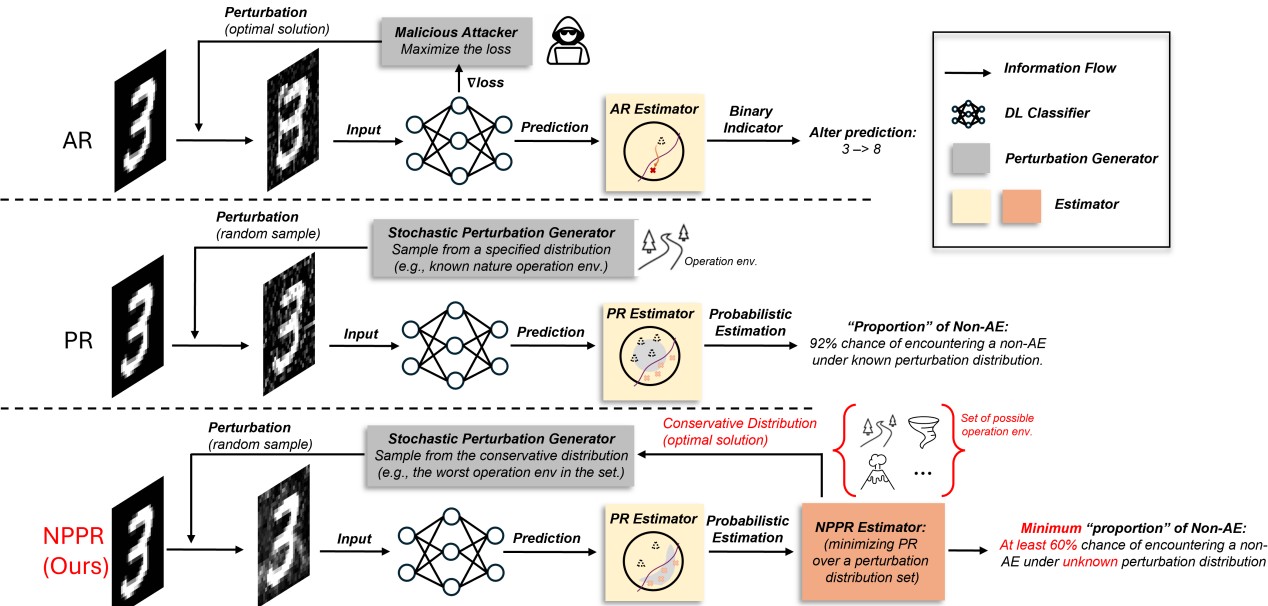

*Figure 1.* **Illustration of robustness estimators under different perturbation assumptions.** AR evaluates worst-case robustness against optimization-based adversarial perturbations, assessing whether such perturbations can alter the prediction. PR estimates robustness by estimating the "proportion" of Non-AEs under known perturbation distribution. The proposed NPPR provides a minimal "proportion" over an admissible perturbation distribution set.

2022; Couellan, 2021; Baluta et al., 2021; Tit et al., 2021; Zhang et al., 2023; Tit et al., 2023; Zhang et al., 2024b) and PR training (Wang et al., 2021a; Robey et al., 2022; Zhang et al., 2024a; 2025; 2026) works. However, in practice, *the perturbation distribution is rarely known a priori.* Although Gaussian or Uniform distributions are commonly used in the literature, these choices are merely illustrative and not intended to represent any universal perturbation distributions. As noted in those PR works, the assessor is expected to specify an appropriate perturbation distribution on a case-by-case basis, supported by justified evidence from the target application—an expectation that is often infeasible in practice.

To bridge the gap arising from the unknown perturbation distribution, we introduce non-parametric[2] probabilistic robustness (NPPR), a more practical PR metric that *does not rely on any predefined perturbation distribution.* Instead, NPPR derives the *conservative* perturbation distribution from a set of distributions (representing, e.g., all possible noise from sensors). This yields the minimum PR over the admissible perturbation distribution set, i.e., providing a lower bound (see Fig. 1).

Specifically, after formally defining NPPR, we develop an NPPR estimator that fits a Gaussian Mixture Model (GMM)

---

[2]The term "non-parametric" follows its usage in statistical modelling, referring to approaches that do not assume a fixed parametric form for the underlying distribution but instead infer it from data.

with Multilayer Perceptron (MLP) heads and bicubic up-sampling to optimize the perturbation distribution from data for the most conservative PR estimates. The estimator considers four dependency scenarios between perturbations, inputs, and labels. We further provide a theoretical analysis of the relationships among AR, PR, and NPPR under these scenarios. Experiments on ResNet18/50, WideResNet50 and VGG16 across CIFAR-10, CIFAR-100, and Tiny ImageNet datasets are conducted to validate our approach.

In summary, our main contributions are as follows:

1. **Formal metric:** We extend the PR concept to a non-parametric setting, formally define the NPPR metric, and theoretically relate it to AR and PR.

2. **Estimator:** We develop an NPPR estimator that learns a conservative perturbation distribution for robustness evaluation under unknown perturbation distribution, with theoretical convergence guarantees.

3. **Open-source repository:** We provide full experimental details and release our implementation publicly at `https://github.com/squarewang2077/nppr`.

## 2. Preliminary and Related Works

Consider a standard classification task, where each input–label pair $(\boldsymbol{x}, y) \in \mathcal{X} \times \mathcal{Y}$ is drawn from an unknown data

distribution $D$ over $\mathcal{X} \times \mathcal{Y}$. $\mathcal{X} \subseteq \mathbb{R}^d$ denotes the input space and $\mathcal{Y} = \{1, 2, \ldots, C\}$ the label space. A hypothesis $h : \mathcal{X} \to \mathcal{Y}$, with $h \in \mathcal{H}$, represents the classifier under study. The training set $S = \{(\boldsymbol{x}_i, y_i)\}_{i=1}^N$ consists of $N$ i.i.d. samples drawn from $D$, and $\ell : \mathcal{H} \times \mathcal{Y} \to \mathbb{R}_+$ denotes the loss function.

We denote perturbations by $\boldsymbol{\varepsilon} \sim \omega$, where $\omega$ is the underlying perturbation distribution, and let $\mathcal{B}$ denote the admissible perturbation budget (e.g., an $L_p$ ball). For convenience in our theoretical analysis, we adopt the notation $\mathfrak{S}$ from prior work on AR (Dohmatob & Bietti, 2022; Wang et al., 2024) to represent our local robustness metric, and $\mathcal{G}$ for corresponding global robustness metric.

## 2.1. Adversarial vs. Probabilistic Robustness

Although definitions of robustness vary across different DL tasks and model types, it generally refers to a model's ability to maintain consistent predictions under small input perturbations. Typically, robustness is defined such that all inputs within a perturbation budget $\mathcal{B}$ yield the same prediction, where $\mathcal{B}$ usually denotes an $L_p$-norm ball of radius $\gamma$ around an input $\boldsymbol{x}$. A perturbed input $\boldsymbol{x}'$ (e.g., obtained by adding noise to $\boldsymbol{x}$) is considered an AE if its predicted label differs from that of $\boldsymbol{x}$. Evaluation methods for robustness are typically based on metrics defined over AEs (Huang et al., 2020; Zühlke & Kudenko, 2024). The formal definitions of AR and PR metrics are introduced in Def. 2.1 and Def. 2.2, respectively.

**Definition 2.1** (Adversarial Robustness). Let $\mathbf{1}_{\mathcal{S}(\boldsymbol{x})}$ be an indicator function that equals 1 if $\mathcal{S}$ holds and 0 otherwise. Given a classifier $h \in \mathcal{H}$, AR around the input $\boldsymbol{x}$ is defined as

$$\mathfrak{S}_{\mathrm{AR}}(\boldsymbol{x}, y) \triangleq 1 - \sup_{\boldsymbol{\varepsilon} \in \mathcal{B}} \mathbf{1}_{h(\boldsymbol{x}+\boldsymbol{\varepsilon}) \neq y}, \qquad (1)$$

where $\mathcal{B} = \{\boldsymbol{\varepsilon} \in \mathbb{R}^d \mid \|\boldsymbol{\varepsilon}\|_p \leq \gamma\}$ denotes the perturbation budget.

**Definition 2.2** (Probabilistic Robustness). Reuse notations in Def. 2.1, and let $\omega(\cdot|\boldsymbol{x})$ denote a perturbation distribution conditioned on $\boldsymbol{x}$, whose support lies within $\mathcal{B}$. Then, the PR of an input–label pair $(\boldsymbol{x}, y)$ is defined as

$$\mathfrak{S}_{\mathrm{PR}}(\boldsymbol{x}, y, \omega) \triangleq \mathbb{E}_{\boldsymbol{\varepsilon} \sim \omega(\cdot|\boldsymbol{x})}\left[\mathbf{1}_{h(\boldsymbol{x}+\boldsymbol{\varepsilon})=y}\right] \qquad (2)$$

*Remark* 2.3 (Input-dependency of perturbations). While Def. 2.2 provides a general formulation of PR in which the perturbation distribution $\omega$ depends on the input $\boldsymbol{x}$, in practice such perturbations *may or may not* depend on the specific input. *Input-dependent perturbations* occur when the characteristics of the noise vary with the input itself. For instance, in computer vision, noise may increase in darker regions of an image, motion blur may depend on the object's velocity. In contrast, *input-independent perturbations* remain statistically identical across all inputs, such

as additive white Gaussian noise, disturbances from constant background vibration or temperature drift that affect all camera inputs equally.

AR[3] captures the "worst-case" scenario in which the generated perturbations yield the AE that maximizes the loss or is closest to the input, typically requiring carefully designed adversarial attack algorithms that often rely on access to model gradients. PR, in contrast, complements AR by estimating the *likelihood* of encountering AEs when perturbations are generated stochastically, ensuring that the risk remains below an acceptable threshold rather than being exactly zero (Webb et al., 2019; Zhang et al., 2023; 2024b; Weng et al., 2019; Zhao, 2026). Intuitively, Def. 2.2 defines PR as the probability that a model's prediction remains unchanged under random perturbations of an input $\boldsymbol{x}$. A "frequentist" interpretation of this probability is the *limiting relative frequency* of perturbations that do not alter the predicted label over infinitely independent trials (Zhang et al., 2024b; Zhao, 2026) (see Fig. 2, panel (b)). We also note that the PR definition in the literature constrains perturbations within a $L_p$-norm ball by enforcing $\|\boldsymbol{\varepsilon}\|_p \leq \gamma$. Equivalently, we assume a noise distribution $\omega$ with zero probability mass outside this budget.

The work (Rice et al., 2021) bridges the gap between PR and AR by introducing a continuum of robustness notions between average-case and worst-case performance. Their work highlights that AR can be viewed as an extreme point of a broader risk-sensitive robustness spectrum, while PR captures less conservative but potentially more operationally relevant notions of robustness. We refer readers to Appendix E.3 for more detailed comparison between our NPPR and the work (Rice et al., 2021).

## 2.2. Probabilistic Robustness Assessment

Recent years have witnessed notable progress in PR research, particularly in terms of its assessment, resulting in the development of a variety of assessment methods across different DL tasks. The earliest study (Webb et al., 2019) to formally define and evaluate PR introduced a black-box statistical estimator based on the Multi-Level Splitting method (Kahn & Harris, 1951), which decomposes the task

---

[3]A related line of work studies AR through randomized or game-theoretic formulations. Meunier et al. (2021) model AEs as a zero-sum game in which both the attacker and the classifier may use mixed strategies, and analyze robustness via mixed Nash equilibria. Other works further investigate randomized defenses against strong attacks (Pinot et al., 2020), game-theoretic formulations of additive adversarial attacks and defenses (Pal & Vidal, 2020), and the role of randomization in adversarially robust classification (Gnecco Heredia et al., 2023). These studies demonstrate that randomized adversaries, randomized classifiers, and equilibrium-based formulations are important alternatives to purely deterministic attack models.

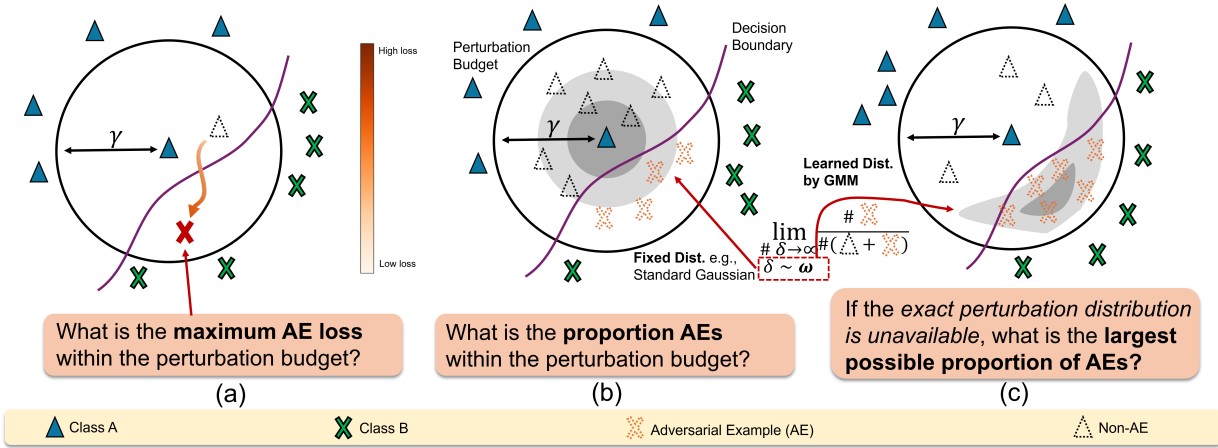

*Figure 2.* Illustration of robustness evaluation paradigms. Panel (a) shows AR, which assesses the worst-case loss within a prescribed perturbation budget. Panel (b) illustrates PR, which measures the "proportion" of non-AEs under a known perturbation distribution. Panel (c) illustrates NPPR, which evaluates the same quantity under a learned conservative perturbation distribution, providing robustness evaluation under unknown perturbation distribution.

of estimating the probability of a rare event into several subproblems and is thus suitable for cases where PR is very high. Later, more efficient white-box estimators were developed in (Tit et al., 2023). Zhang *et al.* (Zhang et al., 2023) further investigated PR under functional perturbations such as color shifts and geometric transformations. In addition, PR has been extended to broader applications such as explainable AI (Huang et al., 2023) and text-to-image models (Zhang et al., 2024b). For a comprehensive overview of PR assessment, readers are referred to (Zhao, 2026). Beyond assessment, several studies (Wang et al., 2021a; Robey et al., 2022; Zhang et al., 2024a; 2025) have focused on developing training methods to improve PR, with their optimization strategies systematically summarized in the recent benchmark study (Zhang et al., 2026).

All aforementioned state-of-the-arts PR formulations rely on a shared assumption that the perturbation noise follows a predefined distribution, which is rarely known in practice. In contrast, our proposed NPPR learns an optimized distribution from data, enabling a conservative evaluation that yields the lowest PR estimate within the admissible distribution set.

## 3. Non-Parametric Probabilistic Robustness

As illustrated in Fig. 2, an inappropriate perturbation distribution (as in existing PR studies that predefine a fixed perturbation distribution; cf. Fig. 2 (b)) may underestimate the true risk. Therefore, an optimized perturbation distribution, adaptively learned from real data, is necessary to overcome the infeasible assumption of a predefined distribution. Accordingly, we formally define our NPPR in Def. 3.1, in which the learned optimized perturbation distribution al-

lows PR to be evaluated more conservatively and accurately.

**Definition 3.1** (Non-parametric PR)**.** Reusing the notation from Def. 2.2, the NPPR of an input–label pair $(\boldsymbol{x}, y)$ is defined as

$$\mathfrak{S}_{\mathrm{NPPR}}(\boldsymbol{x}, y) \triangleq \inf_{\omega \in P_\varepsilon} \mathbb{E}_{\varepsilon \sim \omega(\cdot | \boldsymbol{x})} \left[ \mathbf{1}_{h(\boldsymbol{x}+\varepsilon)=y} \right], \quad (3)$$

where $P_\varepsilon$ is an admissible distribution set with support all lying within the perturbation budget $\mathcal{B}$.

*Remark* 3.2. Similar to Def. 2.2, NPPR also accommodates both input-dependent and input-independent noise distributions (see Remark 2.3). Based on this property, we design different estimators tailored to each type of dependence, enabling a more accurate and conservative PR evaluation.

Building upon the definitions of local robustness for individual input–label pairs (Def. 2.1, 2.2, 3.1), we extend the concept to a global robustness metric that quantifies the overall robustness of a classifier across the entire data distribution.

**Definition 3.3** (Global Robustness)**.** Consider input–label pairs $(\boldsymbol{x}, y) \sim D$, and let $\mathfrak{S}(\boldsymbol{x}, y)$ denote the point-wise robustness metric. The global robustness over the distribution $D$ is defined as

$$\mathcal{G}(D) \triangleq \mathbb{E}_{(\boldsymbol{x},y) \sim D} \left[ \mathfrak{S}(\boldsymbol{x}, y) \right]. \quad (4)$$

For simplicity and to avoid ambiguity, we omit the distribution $D$ from the notation $\mathcal{G}$ and let $\mathcal{G}_{\mathrm{AR}}$, $\mathcal{G}_{\mathrm{PR}}$, and $\mathcal{G}_{\mathrm{NPPR}}$ denote the respective global metrics for AR, PR, and NPPR.

**Proposition 3.4.** *Considering AR, PR, and NPPR as defined in Def. 2.1, 2.2, and 3.1, and binary loss function for AR, let $\mathcal{G}_{\mathrm{AR}}$, $\mathcal{G}_{\mathrm{PR}}$, and $\mathcal{G}_{\mathrm{NPPR}}$ denote their corresponding*

*global robustness metrics. Given a perturbation distribution $\omega \in P_{\varepsilon}$ (either conditional or unconditional) for the perturbation $\varepsilon$, we have*

$$\mathcal{G}_{\text{AR}} \leq \mathcal{G}_{\text{NPPR}} \leq \mathcal{G}_{\text{PR}}. \tag{5}$$

*If we allow $P_{\varepsilon}$ to be unrestricted, representing any distributions (including the Dirac delta measure), then the equality holds,*

$$\mathcal{G}_{\text{AR}} = \mathcal{G}_{\text{NPPR}}. \tag{6}$$

**Proposition 3.5.** *Reuse the condition in Prop. 3.4, and let $\mathcal{G}^{\text{c}}$ and $\mathcal{G}^{\text{u}}$ denote global robustness on conditional and unconditional perturbation distributions for AR and NPPR, respectively. Then we have*

$$\mathcal{G}^{\text{c}} \leq \mathcal{G}^{\text{u}}. \tag{7}$$

Prop. 3.4 shows that the global NPPR metric serves as a more conservative measure compared to PR. Under extreme conditions, NPPR can be as low as AR. Prop. 3.5 compares the global robustness metrics for the conditional and unconditional cases and demonstrates that the conditional case yields lower robustness than the unconditional case. The detailed proofs can be found in Appendix A.1.

# 4. NPPR Estimation

## 4.1. Conservative Distribution via Optimization

We first obtain the conservative distribution by a minimization problem over a set of distributions. To this end, we construct a learnable perturbation distribution that conservatively characterizes admissible perturbations. As illustrated in Fig. 3, the construction consists of a parameterized distribution model that generates perturbations within a prescribed budget, which are then used to evaluate PR via Monte Carlo estimation (Webb et al., 2019). This enables NPPR to assess robustness without assuming a known perturbation distribution. We obtain this conservative distribution by optimization. With this construction in place, we next formalize the objective function under the resulting perturbation distribution.

**Finite-GMM approximation of NPPR** The preceding construction optimizes a learnable perturbation distribution through a finite Gaussian mixture model with a Gumbel–Softmax relaxation. Here, "non-parametric" refers to the metric-level formulation of NPPR, where robustness is defined by optimizing over an admissible class of perturbation distributions rather than a fixed reference distribution. In practice, we approximate this ideal distributional search by restricting the optimization to a finite-$K$ GMM family. Although this restriction may introduce an approximation

gap, it remains more expressive than standard PR with a single prescribed perturbation distribution, since the GMM can model multiple perturbation modes through adaptive mixture weights, means, and covariances. The number of components $K$ therefore controls the trade-off between expressiveness and computational efficiency. A formal relation between the finite-GMM estimator and the ideal NPPR objective is given in Appendix B.

**Objective function** Let $\mathcal{U}$ denote the up-sampling module, which adjusts perturbations to match the input resolution while ensuring that their support lies within the prescribed perturbation budget. Consider a classifier $h$ and a differentiable smooth surrogate loss $\varphi$ that relaxes the hard indicator loss $\mathbf{1}$. Given an input-label pair $(\boldsymbol{x}, y) \sim D$ and perturbations sampled from a GMM, our objective function is

$$\mathcal{L}(\boldsymbol{\phi}) = \mathbb{E}_{(\boldsymbol{x},y)\sim D}\left[\mathbb{E}_{\boldsymbol{\varepsilon}\sim\text{GMM}_{\boldsymbol{\phi}}}\left[\varphi\big(h(\boldsymbol{x} + \mathcal{U}(\boldsymbol{\varepsilon})), y\big)\right]\right] \tag{8}$$

where $\text{GMM}_{\boldsymbol{\phi}} = \text{GMM}(\boldsymbol{\pi}_{\boldsymbol{\phi}}, \boldsymbol{\mu}_{\boldsymbol{\phi}}, \Sigma_{\boldsymbol{\phi}})$ denotes a parameterized Gaussian mixture distribution, whose parameters are governed by $\boldsymbol{\phi}$.

We define the loss function $\varphi$ as the logit margin between the ground-truth class and the most confident non–ground-truth class, with a hyperparameter $\kappa$ controlling the margin scale. Let $\boldsymbol{x}' = \boldsymbol{x} + \mathcal{U}(\boldsymbol{\varepsilon})$ denote the AE. The loss function is

$$\varphi\left(h(\boldsymbol{x}'), y\right) = \text{softplus}\left(h_y(\boldsymbol{x}') - \max_{j \neq y} h_j(\boldsymbol{x}') + \kappa\right) \tag{9}$$

where $\text{softplus}(x) = \log(1 + e^x)$ is lower-bounded by zero, improves its smoothness near the origin, and behaves approximately linearly when the logit gap is large. We experimented with several variants of the loss function, and found that the *softplus* formulation yields the most stable optimization trajectory.

The above objective induces a learnable perturbation distribution for NPPR estimation. To allow the distribution to adapt to input-dependent perturbation patterns, we parameterize it using lightweight MLP heads.

**MLP heads for input-dependent parameterization** We parameterize the perturbation distribution using lightweight MLP heads to enable input-dependent modeling. As illustrated in Fig. 3, the MLP outputs the parameters of a Gaussian mixture model, including the mixture weights $\{\pi_k\}_{k=1}^{K}$, means $\{\mu_k\}_{k=1}^{K}$, and covariances $\{\Sigma_k\}_{k=1}^{K}$. This design allows the perturbation distribution to adapt to different inputs while remaining fully learnable. Additional dependency variants, including input-independent and partially conditioned settings, cf. Fig. 4 in Appendix D.

The first type is **(i) Independence**, where the perturbation distribution is entirely independent of the input–label pairs.

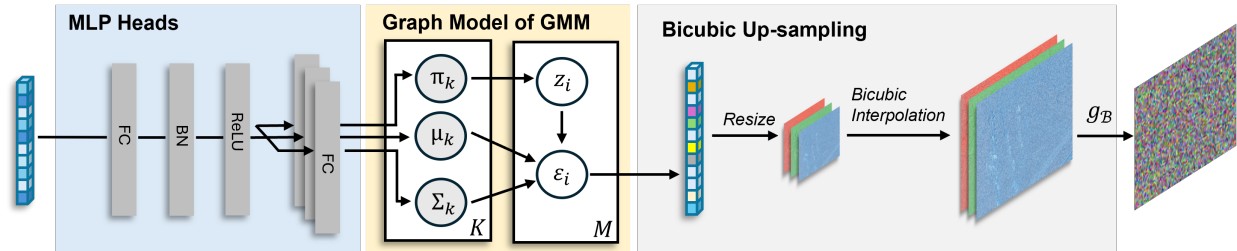

*Figure 3.* **Illustration of the main components for deriving the conservative perturbation distribution.** The construction consists of three components: (i) *MLP heads* that parameterize the perturbation distribution, (ii) a *Gaussian mixture model (GMM)* for sampling latent perturbations, and (iii) *bicubic up-sampling* that maps latent perturbations to the input space. The MLP heads output the parameters of the GMM, from which perturbations are sampled and subsequently up-sampled to the input resolution. The mapping $g_{\mathcal{B}}$ ensures that the resulting perturbations lie within the prescribed perturbation budget.

This corresponds to scenarios such as additive Gaussian noise, external disturbances, and temperature drift, as discussed in Remark 2.3. **(ii) Label dependence:** As shown in Fig. 4 (a), the perturbation distribution depends on the ground-truth labels through a learnable label embedding on mixture proportions. In this case, the noise is characterized by label-specific variations rather than input-dependent ones. **(iii) Input dependence:** The perturbation distribution depends solely on the input features extracted from the classifier. **(iv) Joint dependence:** As illustrated in Fig. 4 (b), the mixture weights $\pi_k$ are linked to ground truth labels, while the means and covariances of the mixture components, $\mu_k$ and $\Sigma_k$, are conditioned on the input features.

**GMM with Gumbel–softmax**   We model the perturbation distribution using a GMM, which represents a distribution as a finite mixture of $K$ Gaussian components. For each input, a latent component index $z_i$ is sampled according to the mixture weights, followed by sampling from the corresponding Gaussian component:

$$z_i \sim \mathrm{Categorical}(\pi_1, \dots, \pi_K), \qquad (10)$$
$$\varepsilon_i \mid z_i \sim \mathcal{N}(\mu_{z_i}, \Sigma_{z_i}). \qquad (11)$$

In our setting, perturbations are not observed explicitly, and the discrete sampling of $z_i$ prevents direct gradient-based optimization. To enable end-to-end training, we employ the Gumbel–Softmax trick (Jang et al., 2017; Maddison et al., 2017), which provides a differentiable relaxation of categorical sampling. Specifically, let $g_k \overset{i.i.d.}{\sim} \mathrm{Gumbel}(0,1)$ and define the relaxed mixture weights

$$w_{\tau,k} = \mathrm{softmax}\left( \frac{\log \pi_{\phi,k} + g_k}{\tau} \right), \quad k \in [K]. \quad (12)$$

The relaxed perturbation is given by $\varepsilon' = \sum_{k=1}^{K} w_{\tau,k} \varepsilon_k$, where $\varepsilon_k \sim \mathcal{N}(\mu_{\phi,k}, \Sigma_{\phi,k})$. This relaxation yields a smooth surrogate objective

$$L_\tau(\phi) = \mathbb{E}_{(\boldsymbol{x},y)} \mathbb{E}_{\boldsymbol{g},\varepsilon_1,\dots,\varepsilon_K} \left[ \varphi \left( h(\boldsymbol{x} + \mathcal{U}(\varepsilon')), y \right) \right], \quad (13)$$

which is differentiable with respect to $\phi$. And the empirical objective on training set $S$ is

$$L_{\tau,S}(\phi) = \frac{1}{N} \sum_{i=1}^{N} \mathbb{E}_{\boldsymbol{g},\varepsilon_1,\dots,\varepsilon_K} \left[ \varphi \left( h(\boldsymbol{x}_i + \mathcal{U}(\varepsilon')), y_i \right) \right]. \qquad (14)$$

Further details of the Gumbel–Softmax construction are provided in Appendix D.1. We next quantify the approximation error introduced by the Gumbel–Softmax relaxation and show that the relaxed objective converges exponentially fast to the original objective as the temperature $\tau$ decreases.

**Lemma 4.1** (Gumbel–Softmax approximation error). *Let $\mathcal{L}(\phi)$ and $\mathcal{L}_\tau(\phi)$ denote the original and Gumbel–Softmax relaxed objectives, respectively. Assume that the loss function $\varphi$ is L-Lipschitz and that the perturbation satisfies $\|\varepsilon\| \leq \gamma$ almost surely. If the underlying mixture admits a unique dominant component, then there exists a constant $C > 0$ such that*

$$\left| \mathcal{L}(\phi) - \mathcal{L}_\tau(\phi) \right| \leq (K-1) L \gamma e^{-\frac{C}{\tau}}. \qquad (15)$$

A more detailed version of this lemma and its proof are provided in Section A.2 of Appendix A.

**Bicubic up-sampling**   Since input images typically reside in high-dimensional spaces, the covariance matrix of the perturbation distribution scales as $\mathcal{O}(d^2)$ with respect to the input dimension $d$, making direct computation infeasible. To address this, we perform perturbation modeling in a lower-dimensional feature space and map the resulting perturbations back to the input space using bicubic interpolation, which is computationally efficient and preserves spatial smoothness (Dong et al., 2016; Keys, 2003). Detailed interpolation formulas are provided in the Appendix C.

To ensure that the support of the perturbation distribution lies within the prescribed $L_p$-norm ball, we apply a scaled $\tanh$ mapping multiplied by the perturbation budget $\gamma$, a common constraint mechanism in the AR literature (Chen et al., 2022; Huang & Zhang, 2020).

## 4.2. Algorithms

---

**Algorithm 1** Conservative Perturbation Distribution via Optimization

---

**Require:** Frozen classifier $h$, feature extractor $f$, training set $S_{\text{train}}$, parametric model $\text{MLP}_\phi$, temperature $\tau$, loss $\varphi$, iterations $T$, step sizes $\{\eta_t\}_{t=1}^T$.
**Ensure:** Trained parameters $\widehat{\phi}$.
1: Initialize $\phi_1$.
2: **for** $t = 1$ **to** $T$ **do**
3:     Sample mini-batch $B_t \subset S_{\text{train}}$.
4:     Keep correctly classified samples $B_t^c \subset B_t$.
5:     Draw auxiliary noise $\zeta$.
6:     Compute stochastic gradient

$$\boldsymbol{g}_t = \nabla_\phi \frac{1}{|B_t^c|} \sum_{(\boldsymbol{x},y) \in B_t^c} \mathbb{E}_{\boldsymbol\zeta}\big[\ell_\tau(\phi_t; (\boldsymbol{x},y), \boldsymbol\zeta)\big].$$

7:     Update $\phi_{t+1} \leftarrow \phi_t - \eta_t \boldsymbol{g}_t$.
8: **end for**
9: **return** $\widehat{\phi} = \phi_{T+1}$.

---

**Algorithm 2** NPPR Evaluation

---

**Require:** Frozen classifier $h$, feature extractor $f$, test set $S_{\text{test}}$, trained model $\text{MLP}_{\widehat{\phi}}$, number of samples $M$.
**Ensure:** Estimated NPPR value $\widehat{\mathcal{G}}_{\text{NPPR}}$.
1: Filter correctly classified test samples:

$$S_{\text{test}}^c = \{(\boldsymbol{x}, y) \in S_{\text{test}} : \arg\max h(\boldsymbol{x}) = y\}.$$

2: Compute

$$\widehat{\mathcal{G}}_{\text{NPPR}} = \frac{1}{|S_{\text{test}}^c| M} \sum_{(\boldsymbol{x},y) \in S_{\text{test}}^c} \sum_{j=1}^M \mathbf{1}_{h(\boldsymbol{x}+\mathcal{U}(\boldsymbol\varepsilon_j))=y},$$

where each $\boldsymbol\varepsilon_j$ is sampled from the GMM parameterized by $(\widehat{\boldsymbol\mu}, \widehat\Sigma, \widehat{\boldsymbol\pi}) = \text{MLP}_{\widehat\phi}(f(\boldsymbol{x}))$.

---

Algorithm 1 trains the NPPR distribution estimator using correctly classified training samples under a frozen classifier. $\ell_\tau(\cdot)$ represents the loss $\varphi(\cdot)$ after reparameterization and auxiliary noise $\zeta$ denotes a collection of noise from standard Gaussian and Gumbel distributions (c.f. Thm. A.4 in Appendix A.2).

Algorithm 2 then estimates NPPR on test set via Monte Carlo sampling from the learned perturbation distributions, with no parameter updates during evaluation. During training, each SGD iteration processes a mini-batch of size $b$ and draws $M$ perturbation samples per input, resulting in a per-iteration complexity of $\mathcal{O}(bM \cdot C_h)$, where $C_h$ denotes the cost of a forward/backward pass through the frozen classifier and the lightweight MLP head. At inference time, NPPR evaluation requires $\mathcal{O}(|S_{\text{test}}^c| M \cdot C_h)$ forward passes, scaling linearly with the number of Monte Carlo samples. We report the detailed wall-clock time costs for both training and inference in Appendix E.1.

## 4.3. Convergence of the Relaxed Objective

**Theorem 4.2** (Nonconvex SGD convergence). *Let $L_{\tau,S}(\phi)$ be an empirical objective and assume it is $\beta$-smooth and bounded below by $L_{\tau,S}^\star \triangleq \inf_\phi L_{\tau,S}(\phi)$. Consider the SGD iterates $\phi_{t+1} = \phi_t - \eta\, \boldsymbol{g}_t$, where $\eta \le 1/\beta$ and $\boldsymbol{g}_t$ is a stochastic gradient estimator satisfying*

$$\mathbb{E}\left[\boldsymbol{g}_t \mid \phi_t\right] = \nabla L_{\tau,S}(\phi_t), \tag{16}$$

$$\mathbb{E}\left[\|\boldsymbol{g}_t - \nabla L_{\tau,S}(\phi_t)\|^2 \mid \phi_t\right] \le \sigma^2 \tag{17}$$

*for some $\sigma^2 < \infty$. Then for any $T \ge 1$,*

$$\frac{1}{T}\sum_{t=0}^{T-1} \mathbb{E}\left[\|\nabla L_{\tau,S}(\phi_t)\|^2\right] \tag{18}$$

$$= \mathcal{O}\left(\frac{L_{\tau,S}(\phi_0) - L_{\tau,S}^\star}{\eta T} + \beta\eta\sigma^2\right), \tag{19}$$

*where $\mathcal{O}(\cdot)$ hides universal numerical constants.*

In our estimator, the variance $\sigma^2$ further decomposes into a Monte-Carlo term and a mini-batch term (c.f. Thm A.4 Appendix A). A theorem with proof is provided in Thm. A.4.

Theorem 4.2 establishes that optimizing the Gumbel–Softmax relaxed objective $L_{\tau,S}$ via SGD enjoys standard convergence guarantees for smooth nonconvex optimization. In particular, the result shows that the expected squared gradient norm converges at the canonical $\mathcal{O}(1/T)$ rate up to a variance-dependent neighborhood, despite the use of Monte Carlo sampling and stochastic mini-batches.

## 5. Experiments

**Experimental setup** We evaluate our method on CIFAR-10, CIFAR-100, and Tiny ImageNet using ResNet18/50, WideResNet50, and VGG16 as base classifiers. The perturbation distribution is modeled by a GMM with varying numbers of components ($K \in \{3, 7, 12\}$) and learned under four conditioning strategies: input-independent, label-dependent, input-dependent, and joint input–label dependence, which together capture a broad range of dependency structures. All evaluations are conducted under an $L_\infty$ perturbation budget with $\gamma \in \{4/255, 8/255, 16/255\}$. The distribution parameters are learned for 50 epochs using Adam with learning rate $5 \times 10^{-4}$. Experiments are implemented in Python 3.10 and PyTorch 2.5.1, and run on two NVIDIA RTX 3090 GPUs. Additional implementation and training details are provided in Appendix D.

*Table 1.* **Performance across datasets and models (%).** Results are reported by taking average with standard deviations in parentheses over 10 runs. $\widehat{\mathcal{G}}_{\mathrm{NPPR}}$ corresponds to the GMM-based estimator. PR is evaluated under Gaussian, Uniform, and Laplace perturbation distributions, while AR is evaluated using PGD, CW for 3 steps with $\alpha = 0.25 \times \gamma$, and standard AutoAttack (AA). Results are obtained under joint dependency with 7 mixture modes and an $L_\infty$ perturbation radius of 16/255. All robustness results are clean-normalized.

| Dataset | Model | Acc. | $\widehat{\mathcal{G}}_{\mathrm{NPPR}}$(Ours) | $\widehat{\mathcal{G}}_{\mathrm{PR_{Gaussian}}}$ | $\widehat{\mathcal{G}}_{\mathrm{PR_{Uniform}}}$ | $\widehat{\mathcal{G}}_{\mathrm{PR_{Laplace}}}$ | $\widehat{\mathcal{G}}_{\mathrm{AR_{PGD}}}$ | $\widehat{\mathcal{G}}_{\mathrm{AR_{CW}}}$ | $\widehat{\mathcal{G}}_{\mathrm{AR_{AA}}}$ |
|---|---|---|---|---|---|---|---|---|---|
| CIFAR10 | ResNet18 | 87.72 | 76.29(0.038) | 95.28(0.030) | 97.21(0.034) | 95.13(0.026) | 3.10(0.084) | 3.30(0.115) | 0.00(0.000) |
| | ResNet50 | 90.75 | 86.84(0.024) | 92.42(0.021) | 94.94(0.020) | 92.23(0.051) | 5.80(0.116) | 6.05(0.128) | 0.00(0.000) |
| | VGG16 | 92.28 | 74.53(0.027) | 89.08(0.033) | 93.36(0.024) | 88.76(0.046) | 0.49(0.019) | 0.31(0.013) | 0.00(0.000) |
| | WRN50 | 91.13 | 81.32(0.030) | 94.20(0.051) | 96.25(0.031) | 94.02(0.020) | 6.94(0.182) | 7.24(0.130) | 0.01(0.000) |
| CIFAR100 | ResNet18 | 63.38 | 58.65(0.018) | 86.51(0.052) | 91.48(0.039) | 86.17(0.042) | 2.45(0.049) | 3.02(0.027) | 0.02(0.000) |
| | ResNet50 | 70.57 | 58.93(0.035) | 80.73(0.047) | 86.78(0.052) | 80.31(0.034) | 3.47(0.079) | 3.69(0.046) | 0.01(0.000) |
| | VGG16 | 71.02 | 52.13(0.015) | 75.86(0.051) | 83.48(0.018) | 75.44(0.067) | 0.98(0.051) | 0.81(0.030) | 0.00(0.000) |
| | WRN50 | 72.02 | 58.88(0.032) | 82.88(0.060) | 88.24(0.034) | 82.58(0.083) | 4.56(0.080) | 4.94(0.101) | 0.01(0.000) |
| TinyImageNet | ResNet18 | 56.99 | 53.20(0.026) | 92.26(0.030) | 95.17(0.025) | 92.07(0.030) | 0.45(0.023) | 0.52(0.021) | 0.00(0.000) |
| | ResNet50 | 70.39 | 64.46(0.044) | 92.09(0.047) | 95.19(0.030) | 91.90(0.025) | 3.90(0.044) | 2.12(0.055) | 0.00(0.000) |
| | VGG16 | 66.95 | 59.01(0.033) | 93.07(0.019) | 95.70(0.040) | 92.87(0.047) | 4.12(0.034) | 0.21(0.014) | 0.00(0.000) |
| | WRN50 | 73.74 | 59.88(0.019) | 93.60(0.013) | 96.19(0.019) | 93.43(0.021) | 4.79(0.039) | 3.18(0.082) | 0.00(0.000) |

*Table 2.* **Ablation settings of ResNet18 on CIFAR-10 under various perturbation dependency settings (%).** $\widehat{\mathcal{G}}_{\mathrm{NPPR}}$ denotes the estimator of the global NPPR metric (c.f. Def. 3.1 & 3.3), while ER represents the entropy ratio of the mixture weights. A smaller ER (c.f. Appendix D.2) implies that the GMM is governed by one dominant component. The lowest value of $\widehat{\mathcal{G}}_{\mathrm{NPPR}}$ is highlighted in bold.

| Config. | Indep. | | Label dep. | | Input Dep. | | Joint Dep. | |
|---|---|---|---|---|---|---|---|---|
| | $\widehat{\mathcal{G}}_{\mathrm{NPPR}}$ | $\mathrm{ER}(\pi)$ | $\widehat{\mathcal{G}}_{\mathrm{NPPR}}$ | $\mathrm{ER}(\pi)$ | $\widehat{\mathcal{G}}_{\mathrm{NPPR}}$ | $\mathrm{ER}(\pi)$ | $\widehat{\mathcal{G}}_{\mathrm{NPPR}}$ | $\mathrm{ER}(\pi)$ |
| Base (K=3) | 74.99 | 0.21 | 75.79 | 99.22 | 85.36 | 57.00 | 79.20 | 99.23 |
| + Learnable | 81.27 | 91.35 | 69.08 | 99.14 | 84.76 | 69.45 | 78.59 | 99.16 |
| + #mode (K=7) | **73.25** | 94.37 | 64.88 | 87.29 | 84.17 | 54.28 | 76.27 | 89.63 |
| + #mode (K=12) | 74.43 | 92.43 | **64.87** | 76.59 | **83.31** | 48.74 | **73.11** | 73.31 |

*Table 3.* **Comparison of AR, PR, and NPPR on CIFAR-10 (ResNet18) under different $L_\infty$ budgets.** Results are reported for $\gamma \in \{4/255, 8/255, 16/255\}$. PR uses a uniform perturbation distribution, AR is evaluated with PGD-10, and NPPR is reported under different dependency settings (K=7).

| Estimator | Dependency | 4/255 | 8/255 | 16/255 |
|---|---|---|---|---|
| $\widehat{\mathcal{G}}_{\mathrm{NPPR}}$ | Indep. | 97.51 | 91.10 | 73.25 |
| | Label dep. | 95.57 | 86.51 | 64.88 |
| | Input dep. | 98.67 | 95.01 | 84.17 |
| | Joint dep. | 96.77 | 90.98 | 76.27 |
| $\widehat{\mathcal{G}}_{\mathrm{PR_{Uniform}}}$ | | 99.71 | 99.30 | 97.26 |
| $\widehat{\mathcal{G}}_{\mathrm{AR_{PGD}}}$ | | 34.91 | 6.33 | 0.05 |

*Table 4.* **NPPR (%) on ResNet18/CIFAR-10 trained and evaluated under different $L_\infty$ radii.** Each row corresponds to a fixed training radius, while each column reports the evaluation result under a different radius, under joint dependency with $K = 7$.

| Train\Eval | 4/255 | 8/255 | 16/255 | 32/255 |
|---|---|---|---|---|
| 4/255 | 96.56 | 91.58 | 79.63 | 59.79 |
| 8/255 | 96.77 | 91.31 | 77.66 | 55.60 |
| 16/255 | 96.75 | 90.99 | 76.27 | 52.49 |

creasing the number of mixture components generally leads to further reductions in $\widehat{\mathcal{G}}_{\mathrm{NPPR}}$, indicating that a richer mixture improves the expressiveness of the learned perturbation distribution. The input-independent variant exhibits a different trend, which we attribute to optimization instability arising from the lack of conditioning information, leading to higher variance in the learned distribution.

### 5.2. Evaluation Across Models and Datasets

Tab. 1 further compares global NPPR, PR (under Gaussian and uniform perturbation distributions), AR (estimated

### 5.1. Ablation Study

Tab. 2 reports an ablation study of the proposed training pipeline on CIFAR-10 with ResNet18. Except for the input-independent setting, all variants yield more conservative robustness estimates, as reflected by lower values of $\widehat{\mathcal{G}}_{\mathrm{NPPR}}$, when a learnable up-sampler is employed. Moreover, in-

using standard first-order attacks as reference), and clean accuracy across different model architectures and datasets. As expected, AR yields the most pessimistic robustness estimates, while PR under fixed perturbation distributions produces the most optimistic ones. Notably, NPPR consistently lies between AR and PR across all configurations, empirically validating Proposition 3.4 and demonstrating that NPPR provides a more conservative yet stable robustness evaluation. In addition, NPPR degrades as the classification task becomes more challenging for a fixed architecture, e.g., $\widehat{\mathcal{G}}_{\mathrm{NPPR}}$ for ResNet18 decreases from 76.29 on CIFAR-10 to 58.65 on CIFAR-100 and 53.20 on TinyImageNet. We also compute the wall time for our NPPR estimator in Tab. 8. Across different datasets, we find that input dimensionality and model complexity are the primary factors affecting efficiency.

To further assess this efficiency aspect, Tab. 5 evaluates NPPR with different feature extractors for the GMM head, suggesting that NPPR does not necessarily require the same or largest backbone as the target classifier. Smaller or different feature extractors can still provide competitive estimates, offering a practical way to reduce the additional fitting cost.

### 5.3. Cross-Radius Analysis

Tab. 3 reports NPPR, PR, and AR under different perturbation radii. Across all settings, smaller perturbation budgets lead to higher robustness estimates, reflecting fewer admissible AEs within a more constrained region.

Tab. 4 reports NPPR scores on ResNet18/CIFAR-10 when the estimator is trained at different radii and evaluated across multiple radii. Two consistent patterns can be observed. First, for a fixed training radius, the NPPR score decreases monotonically as the evaluation radius increases, which matches the expected behavior under a larger perturbation budget. Second, estimators trained at different radii remain reasonably stable across evaluation radii. In particular, training with a larger radius tends to yield slightly more conservative estimates at larger evaluation radii. For example, when evaluated at $32/255$, training at $16/255$ gives an NPPR score of 52.49, compared with 59.79 when training at $4/255$. More cross-radius experimental results are provided in Appendix E.2.

## 6. Conclusion

To address the limitations of PR evaluation caused by reliance on *predefined* perturbation distributions, we introduce *Non-Parametric Probabilistic Robustness* (NPPR). NPPR provides a *conservative* robustness evaluation by estimating PR under an optimized perturbation distribution (from data) that minimizes PR over all admissible distributions, eliminating the need to assume a fixed perturbation distribu-

*Table 5.* **NPPR (%) on CIFAR-10 under $L_\infty(16/255)$ using different feature extractors for the GMM head.** Rows denote the feature extractor used to train the estimator, while columns denote the target classifier being evaluated. Results are obtained under the joint dependency setting with $K = 7$.

| Feat. | ResNet18 | ResNet50 | VGG16 | WRN50 |
|---|---|---|---|---|
| ResNet18 | 76.27 | 80.12 | 74.99 | 77.26 |
| ResNet50 | 86.41 | 86.81 | 86.03 | 86.09 |
| VGG16 | 76.60 | 80.66 | 74.47 | 77.84 |
| WRN50 | 82.40 | 84.21 | 79.79 | 81.32 |

tion a priori. We solve the NPPR estimation problem via a GMM-based implementation and analyze how different dependency structures between inputs and perturbations affect robustness estimates, observing that stronger dependency consistently leads to more conservative NPPR values. Experiments across multiple datasets and model architectures demonstrate that our NPPR approach is both effective and efficient. We further provide theoretical analysis clarifying the relationships among AR, PR, and NPPR.

We note that NPPR *is not intended to design new attacks* (as other common robustness studies). Instead, NPPR serves as a conservative robustness estimator from the *assessor's* perspective, aiming to answer the question: "*if the input is subject to some stochastic perturbations (e.g., arising from background noise, sensor vibrations, or even unsophisticated random attacks), and the exact perturbation distribution is unknown, how robust can the model be at least?*". Thus, NPPR does not replace AR or PR, rather, it complements them by filling the gap created by unknown perturbation distributions when doing assessment. We believe NPPR represents an important stepping stone toward making PR more implementable in real-world applications.

## Acknowledgements

SK and XZ have received funding from the European Union's EU Framework Program for Research and Innovation Europe Horizon (grant agreement No 101202457). ZW's contribution is supported by the EU funded SYNERGIES project (grant agreement No 101146542). XZ's contribution is also supported by the UK EPSRC New Investigator Award [EP/Z536568/1]. SK's contribution is supported by the UKRI Future Leaders Fellowship Grant [MR/S035176/1]. YZ's contribution is supported by China Scholarship Council.

Views and opinions expressed are those of the authors only and do not necessarily reflect those of the European Union or European Research Executive Agency (REA). Neither the European Union nor the granting authority can be held responsible for them.

## Impact Statement

This paper presents work whose goal is to advance the field of machine learning, particularly in robustness evaluation. There are many potential societal consequences of our work, none of which we feel must be specifically highlighted here.

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

# A. Omitted Proofs

## A.1. Relationships among AR, PR and NPPR

To facilitate readability, we restate Propositions 3.4 and 3.5, followed by their proofs in order.

**Proposition A.1.** *Considering AR, PR, and NPPR as defined in Def. 2.1, 2.2, and 3.1, and binary loss function for AR, let $\mathcal{G}_{\mathrm{AR}}$, $\mathcal{G}_{\mathrm{PR}}$, and $\mathcal{G}_{\mathrm{NPPR}}$ denote their corresponding global robustness metrics. Given a perturbation distribution $\omega \in P_{\varepsilon}$ (either conditional or unconditional) for the perturbation $\varepsilon$, we have*

$$\mathcal{G}_{\mathrm{AR}} \leq \mathcal{G}_{\mathrm{NPPR}} \leq \mathcal{G}_{\mathrm{PR}}. \tag{20}$$

*If we allow $P_{\varepsilon}$ to be unrestricted, representing any family of distributions (including the Dirac delta measure), then the equality holds,*

$$\mathcal{G}_{\mathrm{AR}} = \mathcal{G}_{\mathrm{NPPR}}. \tag{21}$$

*If $P_{\varepsilon}$ includes only continuous distributions and the set of all adversarial perturbations is set of measure zero within $\mathcal{B}$, then the following strict inequality holds,*

$$\mathcal{G}_{\mathrm{AR}} < \mathcal{G}_{\mathrm{NPPR}}. \tag{22}$$

*Proof.* Here we provide the proof of Proposition 3.4. The inequality $\mathcal{G}_{\mathrm{NPPR}} \leq \mathcal{G}_{\mathrm{PR}}$ follows directly from Definition 3.1. Thus, it remains to establish the inequality between AR and NPPR, namely $\mathcal{G}_{\mathrm{AR}} \leq \mathcal{G}_{\mathrm{NPPR}}$. We prove this inequality by showing that $\mathcal{G}_{\mathrm{AR}} = \mathcal{G}_{\mathrm{NPPR}}$ when no restrictions are imposed on $P_{\varepsilon}$, and for some specific restrictions imposed on $P_{\varepsilon}$, there exists $\mathcal{G}_{\mathrm{AR}} < \mathcal{G}_{\mathrm{NPPR}}$.

We begin with the equality. To this end, we first establish that $\mathcal{G}_{\mathrm{AR}} \geq \mathcal{G}_{\mathrm{NPPR}}$, and then show the reverse inequality $\mathcal{G}_{\mathrm{AR}} \leq \mathcal{G}_{\mathrm{NPPR}}$. Considering the conditional case and a binary loss function, let

$$\varepsilon^{\star} \in \arg\sup_{\varepsilon \in \mathcal{B}} \mathbf{1}_{h(x+\varepsilon) \neq y} \tag{23}$$

be the adversarial perturbation, and

$$\omega^{\star}(\cdot|x, y) = \arg\inf_{\omega \in P_{\varepsilon}} \mathbb{E}_{\varepsilon \sim \omega(\cdot|x,y)}\left[\mathbf{1}_{h(x+\varepsilon)=y}\right] \tag{24}$$

be optimal perturbation distribution of NPPR. Now, considering Dirac delta measure $\delta_{\varepsilon^{\star}} \in P_{\varepsilon}$, we have

$$\mathcal{G}_{\mathrm{AR}} = \mathbb{E}_D\left[\mathbf{1}_{h(x+\varepsilon^{\star})=y}\right] \tag{25}$$

$$= \mathbb{E}_D\left[\mathbb{E}_{\delta_{\varepsilon^{\star}}}\left[\mathbf{1}_{h(x+\varepsilon)=y}\right]\right] \tag{26}$$

$$\geq \mathbb{E}_D\left[\inf_{\omega} \mathbb{E}_{\omega}\left[\mathbf{1}_{h(x+\varepsilon)=y}\right]\right] \tag{27}$$

$$= \mathcal{G}_{\mathrm{NPPR}} \tag{28}$$

Inversely, we have

$$\mathcal{G}_{\mathrm{NPPR}} = \mathbb{E}_D\left[\mathbb{E}_{\omega^{\star}}\left[\mathbf{1}_{h(x+\varepsilon)=y}\right]\right], \tag{29}$$

If $\mathbb{E}_{\omega^{\star}}\left[\mathbf{1}_{h(x+\varepsilon)=y}\right] < 1$, there must exsit at least one $\varepsilon^{\star} \in \mathcal{B}$, such that $h(x + \varepsilon^{\star}) \neq y$, therefore $\mathbf{1}_{h(x+\varepsilon^{\star})=y} = 0$, hence

$$\mathbb{E}_D\left[\mathbf{1}_{h(x+\varepsilon^{\star})=y}\right] \leq \mathbb{E}_D\left[\mathbb{E}_{\omega^{\star}}\left[\mathbf{1}_{h(x+\varepsilon)=y}\right]\right]. \tag{30}$$

Therefore, if we do not restrict $P_{\varepsilon}$, we have $\mathcal{G}_{\mathrm{NPPR}} = \mathcal{G}_{\mathrm{AR}}$. In case of $P_{\varepsilon}$ represents continuous distributions, and there only exists distinct AEs, $\varepsilon^{\star}$, such that $\mathbf{1}_{h(x+\varepsilon^{\star})=y} = 0$ with probability of zero, then $\forall \omega \in P_{\varepsilon}$ we have $\mathbb{E}_{\omega}[\mathbf{1}_{h(x+\varepsilon)=y}] = 1$. Hence, we have

$$\mathbb{E}_D\left[\mathbf{1}_{h(x+\varepsilon^{\star})=y}\right] < \mathbb{E}_D\left[\mathbb{E}_{\omega^{\star}}\left[\mathbf{1}_{h(x+\varepsilon)=y}\right]\right]. \tag{31}$$

In the unconditional case, instead of the usual input-dependent PGD, we consider Universal Adversarial Attacks (UAEs) (Chaubey et al., 2020). The following part will show that the usual input-dependent attack yields a lower value than that of the UAE. Let

$$\omega^{\star} = \arg\inf_{\omega \in P_{\varepsilon}} \mathbb{E}_D\left[\mathbb{E}_{\varepsilon \sim \omega}\left[\mathbf{1}_{h(x+\varepsilon)=y}\right]\right] \tag{32}$$

$$= \arg\inf_{\omega \in P_{\varepsilon}} \mathbb{E}_{\varepsilon \sim \omega}\left[\mathbb{E}_D\left[\mathbf{1}_{h(x+\varepsilon)=y}\right]\right] \tag{33}$$

The two expectations is exchangeable since $\omega$ is independent of input-label pairs. Let

$$\varepsilon^{\star} \in \arg\sup_{\varepsilon \in \mathcal{B}} \mathbb{E}_D\left[\mathbf{1}_{h(x+\varepsilon) \neq y}\right] \tag{34}$$

be an UAE. Similarly, we have

$$\mathcal{G}_{\mathrm{AR}} = \mathbb{E}_D\left[\mathbf{1}_{h(x+\varepsilon^{\star})=y}\right] \tag{35}$$

$$= \mathbb{E}_{\delta_{\varepsilon^{\star}}}\left[\mathbb{E}_D\left[\mathbf{1}_{h(x+\varepsilon)=y}\right]\right] \tag{36}$$

$$\geq \inf_{\omega} \mathbb{E}_{\omega}\left[\mathbb{E}_D\left[\mathbf{1}_{h(x+\varepsilon)=y}\right]\right] \tag{37}$$

$$= \mathcal{G}_{\mathrm{NPPR}} \tag{38}$$

Now, we show that $\mathcal{G}_{\mathrm{AR}} \leq \mathcal{G}_{\mathrm{NPPR}}$. We have

$$\mathcal{G}_{\mathrm{NPPR}} = \mathbb{E}_D\left[\mathbb{E}_{\omega^{\star}}\left[\mathbf{1}_{h(x+\varepsilon)=y}\right]\right] \tag{39}$$

$$= \mathbb{E}_{\omega^{\star}}\left[\mathbb{E}_D\left[\mathbf{1}_{h(x+\varepsilon)=y}\right]\right]. \tag{40}$$

where $\omega^\star$ is the optimal distribution derived from NPPR, and for any distribution $\omega$, we have

$$\mathbb{E}_\omega \left[ \mathbb{E}_D \left[ \mathbf{1}_{h(\boldsymbol{x}+\boldsymbol{\varepsilon})=y} \right] \right] \geq \inf_{\boldsymbol{\varepsilon}} \mathbb{E}_D \left[ \mathbf{1}_{h(\boldsymbol{x}+\boldsymbol{\varepsilon})=y} \right]. \quad (41)$$

Hence we have $\mathcal{G}_{\text{NPPR}} \geq \mathcal{G}_{\text{AR}}$. The proof of inequality $\mathcal{G}_{\text{NPPR}} > \mathcal{G}_{\text{AR}}$ is the same as the conditional case. $\square$

Now, we prove the Prop. 3.5.

**Proposition A.2.** *Reuse the condition in Prop. 3.4, and let $\mathcal{G}^c$ and $\mathcal{G}^u$ denote global robustness on conditional and unconditional perturbation distributions for AR, and NPPR, respectively. Then we have*

$$\mathcal{G}^c \leq \mathcal{G}^u. \quad (42)$$

*Proof.* We first prove the AR case and consider the UAE as the unconditional case of AR. We have

$$\mathcal{G}_{\text{AR}}^c = 1 - \mathbb{E}_D \left[ \sup_{\boldsymbol{\varepsilon}} \mathbf{1}_{h(\boldsymbol{x}+\boldsymbol{\varepsilon})\neq y} \right] \quad (43)$$

$$= \mathbb{E}_D \left[ \inf_{\boldsymbol{\varepsilon}} \mathbf{1}_{h(\boldsymbol{x}+\boldsymbol{\varepsilon})=y} \right]. \quad (44)$$

Since $\forall \boldsymbol{\varepsilon}_0 \in \mathcal{B}$ and $(\boldsymbol{x},y) \in \mathcal{X} \times \mathcal{Y}, \inf_{\boldsymbol{\varepsilon}} \mathbf{1}_{h(\boldsymbol{x}+\boldsymbol{\varepsilon})=y} \leq \mathbf{1}_{h(\boldsymbol{x}+\boldsymbol{\varepsilon}_0)=y}$, hence $\forall \boldsymbol{\varepsilon}_0 \in \mathcal{B}$

$$\mathbb{E}_D \left[ \inf_{\boldsymbol{\varepsilon}} \mathbf{1}_{h(\boldsymbol{x}+\boldsymbol{\varepsilon})=y} \right] \leq \mathbb{E}_D \left[ \mathbf{1}_{h(\boldsymbol{x}+\boldsymbol{\varepsilon}_0)=y} \right]. \quad (45)$$

Therefore,

$$\mathbb{E}_D \left[ \inf_{\boldsymbol{\varepsilon}} \mathbf{1}_{h(\boldsymbol{x}+\boldsymbol{\varepsilon})=y} \right] \leq \inf_{\boldsymbol{\varepsilon}} \mathbb{E}_D \left[ \mathbf{1}_{h(\boldsymbol{x}+\boldsymbol{\varepsilon})=y} \right] = \mathcal{G}_{\text{AR}}^u. \quad (46)$$

Following the same logic, we show that.

$$\mathcal{G}_{\text{NPPR}}^c = \mathbb{E}_D \left[ \inf_{\omega} \mathbb{E}_\omega \left[ \mathbf{1}_{h(\boldsymbol{x}+\boldsymbol{\varepsilon})=y} \right] \right] \quad (47)$$

$$\leq \inf_{\omega} \mathbb{E}_D \left[ \mathbb{E}_\omega \left[ \mathbf{1}_{h(\boldsymbol{x}+\boldsymbol{\varepsilon})=y} \right] \right] \quad (48)$$

$$= \mathcal{G}_{\text{NPPR}}^u. \quad (49)$$

$\square$

## A.2. Convergence Analysis

**Lemma A.3** (Approximation error of Gumbel–Softmax relaxation). *Let $\mathcal{L}(\boldsymbol{\phi})$ and $\mathcal{L}_\tau(\boldsymbol{\phi})$ denote the objectives defined in Eq. 8 and Eq. 13, respectively. Assume that the loss function $\varphi$ is L-Lipschitz. Suppose the perturbation budget satisfies $\|\boldsymbol{\varepsilon}\| \leq \gamma$ almost surely. Assume that for the Gumbel–Max representation of the categorical variable, there exists a unique maximizer*

$$k^\star = \arg \max_{k \in [K]} (\log \pi_k + g_k), \quad (50)$$

*and define the margin*

$$\Delta \triangleq \min_{k \neq k^\star} \left[ (\log \pi_{k^\star} + g_{k^\star}) - (\log \pi_k + g_k) \right] > 0. \quad (51)$$

*Then there exists a constant $C > 0$ such that*

$$\left| \mathcal{L}(\boldsymbol{\phi}) - \mathcal{L}_\tau(\boldsymbol{\phi}) \right| \leq (K-1) L \gamma e^{-\frac{C}{\tau}}. \quad (52)$$

*Proof.* For simplicity and concise of our analysis, assume a fixed bicubic up-sampler $\mathcal{U}$ and denote $G(\boldsymbol{\varepsilon}; \boldsymbol{x}, y) = \varphi(h(\boldsymbol{x} + \mathcal{U}(\boldsymbol{\varepsilon})), y)$ and assume it is $L$-Lipschitz w.r.t. $\boldsymbol{\varepsilon}$ in a norm space, and let $\boldsymbol{\pi} = (\pi_1, ... \pi_K), \pi_k \geq 0, \sum_{k=1}^K \pi = 1$ be weights related to GMM, $\text{Cat}(\boldsymbol{\pi})$ be category distribution with probability of category $k$ be $\pi_k$, $\mathcal{N}(\boldsymbol{\mu}_k, \Sigma_k)$ be Gaussian, hence our objective function becomes

$$\mathcal{L}(\boldsymbol{\phi}) = \mathbb{E}_{(\boldsymbol{x},y) \sim D} \mathbb{E}_{\boldsymbol{\varepsilon} \sim \text{GMM}_\phi} [G(\boldsymbol{\varepsilon}; \boldsymbol{x}, y)] \quad (53)$$

$$= \mathbb{E}_{(\boldsymbol{x},y) \sim D} \mathbb{E}_{k \sim \text{Cat}(\boldsymbol{\pi}), \boldsymbol{\varepsilon} \sim \mathcal{N}(\boldsymbol{\mu}_k, \Sigma_k)} [G(\boldsymbol{\varepsilon}; \boldsymbol{x}, y)] \quad (54)$$

Since $k \sim \text{Cat}(\boldsymbol{\pi})$ is equivalent to Gumbel-max, Let $\boldsymbol{g} = (g_1, \ldots, g_K)$ be a Gumbel distributed r.v. such that $g_k \overset{i.i.d.}{\sim}$ Gumbel$(0, 1)$, we have

$$k^\star = \arg \max_{k \in [K]} (\log \pi_k + g_k) \sim \text{Cat}(\boldsymbol{\pi}) \quad (55)$$

Therefore,

$$\mathcal{L}(\boldsymbol{\phi}) = \mathbb{E}_{(\boldsymbol{x},y)} \mathbb{E}_{\boldsymbol{g}} \mathbb{E}_{\boldsymbol{\varepsilon} \sim \mathcal{N}(\boldsymbol{\mu}_{k^\star}, \Sigma_{k^\star})} [G(\boldsymbol{\varepsilon}; \boldsymbol{x}, y)] \quad (56)$$

Gumbel-softmax trick smooth the category random variable with softmax function, such that $\boldsymbol{\varepsilon}$ is not drawing from $\mathcal{N}(\boldsymbol{\mu}_{k^\star}, \Sigma_{k^\star})$, instead, it draws from a weighted sum of $K$ Gaussian with a temperature $\tau$. The weight is given by

$$w_{\tau,k} = \text{softmax} \left( \frac{\log \pi_k + g_k}{\tau} \right) \text{ for } k \in [K] \quad (57)$$

and the corresponding $\boldsymbol{\varepsilon}'$ is

$$\boldsymbol{\varepsilon}' = \sum_{k=1}^K w_{\tau,k} \boldsymbol{\varepsilon}_k \quad (58)$$

The $L_\tau(\boldsymbol{\phi})$ is hence constructed as

$$L_\tau(\boldsymbol{\phi}) = \mathbb{E}_{(\boldsymbol{x},y)} \mathbb{E}_{\boldsymbol{g}} \mathbb{E}_{\boldsymbol{\varepsilon}_1,...,\boldsymbol{\varepsilon}_K} [G(\boldsymbol{\varepsilon}'; \boldsymbol{x}, y)] \quad (59)$$

Hence,

$$|L(\boldsymbol{\phi}) - L_\tau(\boldsymbol{\phi})| \quad (60)$$

$$\leq \mathbb{E}_{(\boldsymbol{x},y)} \mathbb{E}_{\boldsymbol{g}} \mathbb{E}_{\boldsymbol{\varepsilon}_1,...,\boldsymbol{\varepsilon}_K} |G(\boldsymbol{\varepsilon}_{\boldsymbol{k}^\star}; \boldsymbol{x}, y) - G(\boldsymbol{\varepsilon}'; \boldsymbol{x}, y)| \quad (61)$$

$$\leq L \mathbb{E}_{(\boldsymbol{x},y)} \mathbb{E}_{\boldsymbol{g}} \mathbb{E}_{\boldsymbol{\varepsilon}_1,...,\boldsymbol{\varepsilon}_K} \|\boldsymbol{\varepsilon}_{\boldsymbol{k}^\star} - \boldsymbol{\varepsilon}'\| \quad (62)$$

where

$$\varepsilon_{k^\star} - \varepsilon' = \sum_{k \neq k^\star} w_\tau \varepsilon_k \tag{63}$$

And since each $\varepsilon_k$ is bounded $\gamma$, we have

$$\|\varepsilon_{k^\star} - \varepsilon'\| \leq \gamma(1 - w_{\tau,k^\star}). \tag{64}$$

Denote $s_k = \log \pi_k + g_k$ and $\Delta_k = s_{k^\star} - s_k \geq 0$. Then the Gumbel–Softmax weight corresponding to the maximal logit satisfies

$$w_{\tau,k^\star} = \frac{e^{s_{k^\star}/\tau}}{\sum_j e^{s_j/\tau}} = \frac{1}{1 + \sum_{k \neq k^\star} e^{-\Delta_k/\tau}}. \tag{65}$$

Consequently,

$$1 - w_{\tau,k^\star} = \frac{\sum_{k \neq k^\star} e^{-\Delta_k/\tau}}{1 + \sum_{k \neq k^\star} e^{-\Delta_k/\tau}} \tag{66}$$

$$\leq \sum_{k \neq k^\star} e^{-\Delta_k/\tau}. \tag{67}$$

Let $k_m \triangleq \arg\min_{k \neq k^\star} \Delta_k$ which yields

$$|L(\phi) - L_\tau(\phi)| \leq (K-1)L\gamma \mathbb{E}_g e^{-\Delta_{k_m}/\tau} \tag{68}$$

$$\leq (K-1)L\gamma e^{-(\log \pi_{k^\star} - \log \pi_{k_m})/\tau} \tag{69}$$

$\square$

**Theorem A.4** (Nonconvex SGD convergence for $L_{\tau,S}(\phi)$). *Let $z = (x, y)$, the objective function in Eq. 13 can be reparameterized as*

$$L_\tau(\phi) = \mathbb{E}_{(x,y)\sim D} \mathbb{E}_\zeta \Big[ \ell_\tau(\phi; z, \zeta) \Big], \tag{70}$$

*where $\zeta$ denotes the auxiliary noise induced by the Gumbel–Softmax and Gaussian reparameterizations. Consider the empirical objective*

$$L_{\tau,S}(\phi) \triangleq \frac{1}{N} \sum_{i=1}^N \mathbb{E}_\zeta \big[ \ell_\tau(\phi; z_i, \zeta) \big], \tag{71}$$

*Assume:*

*(A0)* (Regularity / differentiation under expectation) *For all $\phi$ and $z$,*

$$\nabla_\phi \mathbb{E}_\zeta \left[ \ell_\tau(\phi; z, \zeta) \right] = \mathbb{E}_\zeta \left[ \nabla_\phi \ell_\tau(\phi; z, \zeta) \right]. \tag{72}$$

*(A1)* (Bounded noise-induced variance) *There exists $\sigma_\zeta^2 < \infty$ such that for all $\phi$ and $z$,*

$$\mathbb{E}_\zeta \left[ \big\| \nabla_\phi \ell_\tau(\phi; z, \zeta) - m(\phi; z) \big\|^2 \right] \leq \sigma_\zeta^2, \tag{73}$$

*where $m(\phi; z) \triangleq \mathbb{E}_\zeta \left[ \nabla_\phi \ell_\tau(\phi; z, \zeta) \right]$.*

*(A2)* (Bounded data variance) *There exists $\sigma_S^2 < \infty$ such that for all $\phi$,*

$$\mathbb{E}_{z \sim \text{Unif}(S)} \left[ \| m(\phi; z) - \nabla L_{\tau,S}(\phi) \|^2 \right] \leq \sigma_S^2. \tag{74}$$

*(A3)* (Smoothness and lower boundedness) *The function $L_{\tau,S}(\phi)$ is $\beta$-smooth and bounded below by $L_{\tau,S}^\star \triangleq \inf_\phi L_{\tau,S}(\phi)$.*

*Consider SGD with step size $\eta \leq 1/\beta$, the update of the parameter is*

$$\phi_{t+1} = \phi_t - \eta g_t \tag{75}$$

$$g_t \triangleq \frac{1}{bM} \sum_{z \in B_t} \sum_{j=1}^M \nabla_\phi \ell_\tau(\phi_t; z, \zeta_j), \tag{76}$$

*where $B_t$ is a uniform mini-batch of size $b$ and $\{\zeta_j\}_{j=1}^M$ are i.i.d. Then for any $T \geq 1$,*

$$\frac{1}{T} \sum_{t=0}^{T-1} \mathbb{E}\left[ \| \nabla L_{\tau,S}(\phi_t) \|^2 \right] \tag{77}$$

$$= \mathcal{O}\left( \frac{L_{\tau,S}(\phi_0) - L_{\tau,S}^\star}{\eta T} + \beta\eta\left( \frac{\sigma_\zeta^2}{bM} + \frac{\sigma_S^2}{b} \right) \right). \tag{78}$$

*where $\mathcal{O}(\cdot)$ hides universal numerical constants.*

*Proof.* We prove our theorem from following steps:

**Step 1: Reparameterization of $L_\tau$** Reuse the notation of proof in Lem. A.3, we have

$$L_\tau(\phi) = \mathbb{E}_{(x,y)\sim D} \mathbb{E}_g \mathbb{E}_{\varepsilon_1,\dots,\varepsilon_K} \big[ G(\varepsilon'; x, y) \big], \tag{79}$$

where

$$G(\varepsilon; x, y) = \varphi(h(x + \mathcal{U}(\varepsilon)), y) \tag{80}$$

$$\varepsilon' = \sum_{k=1}^K w_{\tau,k} \varepsilon_k. \tag{81}$$

The relaxed mixture weights are given by the Gumbel–Softmax construction

$$w_{\tau,k} = \frac{e^{(\log \pi_k + g_k)/\tau}}{\sum_{j=1}^K e^{(\log \pi_j + g_j)/\tau}} \tag{82}$$

$$\text{where } g_k \overset{i.i.d.}{\sim} \text{Gumbel}(0, 1). \tag{83}$$

Let $\xi_k \sim \mathcal{N}(0, I), \forall k \in [K]$, we reparameterize each Gaussian component as

$$\varepsilon_k = \mu_k + A_k \xi_k, \tag{84}$$

where $A_k A_k^T = \Sigma_k$, denoting the Cholesky decomposition. Hence, we have our reparameterized results as

$$\varepsilon'(\phi; \boldsymbol{x}, y, \boldsymbol{g}, \boldsymbol{\xi}) = \sum_{k=1}^{K} w_{\tau,k}(\phi; \boldsymbol{x}, y, \boldsymbol{g}) \boldsymbol{\mu}_k(\phi; \boldsymbol{x}, y) \quad (85)$$

$$+ \sum_{k=1}^{K} w_{\tau,k}(\phi; \boldsymbol{x}, y, \boldsymbol{g}) A_k(\phi; \boldsymbol{x}, y) \boldsymbol{\xi}_k, \quad (86)$$

where $\boldsymbol{\xi} \triangleq (\boldsymbol{\xi}_1, \ldots, \boldsymbol{\xi}_K), \boldsymbol{g} = (g_1, \ldots, g_K)$. Defining the auxiliary noise variable

$$\boldsymbol{\zeta} \triangleq (\boldsymbol{g}, \boldsymbol{\xi}), \quad (87)$$

For notational simplicity, we drop the $(\boldsymbol{x}, y)$ from $G(\varepsilon; \boldsymbol{x}, y)$ and we equivalently rewrite the relaxed objective as

$$L_\tau(\phi) = \mathbb{E}_{(\boldsymbol{x}, y) \sim D} \mathbb{E}_{\boldsymbol{\zeta}} \Big[ G\big(\varepsilon'(\phi; \boldsymbol{x}, y, \boldsymbol{\zeta})\big) \Big]. \quad (88)$$

Under standard regularity conditions, the gradient of $L_\tau$ admits the form

$$\nabla_\phi L_\tau(\phi) = \mathbb{E}_{(\boldsymbol{x}, y) \sim D} \mathbb{E}_{\boldsymbol{\zeta}} \Big[ \nabla_\phi G\big(\varepsilon'(\phi; \boldsymbol{x}, y, \boldsymbol{\zeta})\big) \Big], \quad (89)$$

which forms the basis of the stochastic gradient updates used in Algorithm 1. For convenience, let $\boldsymbol{z} = (\boldsymbol{x}, y)$ and

$$\ell_\tau(\phi; \boldsymbol{z}, \boldsymbol{\zeta}) \triangleq G\big(\varepsilon'(\phi; \boldsymbol{z}, \boldsymbol{\zeta})\big), \quad (90)$$

the empirical objective becomes

$$L_{\tau,S}(\phi) \triangleq \frac{1}{N} \sum_{i=1}^{N} \mathbb{E}_{\boldsymbol{\zeta}} \big[ \ell_\tau(\phi; \boldsymbol{z}_i, \boldsymbol{\zeta}) \big]. \quad (91)$$

**Step 2: Stochastic Gradient Estimator** We first show that our gradient estimator is unbiased. Consider the empirical objective, At iteration $t$, let $B_t$ be a mini-batch of size $b$ sampled uniformly from the training set, and let $\{\boldsymbol{\zeta}_j\}_{j=1}^{M}$ denote independent realizations of the auxiliary noise. The stochastic gradient estimator used by Algorithm 1 is given by

$$\boldsymbol{g}_t \triangleq \frac{1}{bM} \sum_{\boldsymbol{z} \in B_t} \sum_{j=1}^{M} \nabla_\phi \ell_\tau(\phi_t; \boldsymbol{z}, \boldsymbol{\zeta}_j). \quad (92)$$

Under standard regularity conditions (dominated convergence) that justify interchanging gradient and expectation, the estimator $\boldsymbol{g}_t$ is unbiased in the sense that (For clarity, we temporarily drop the condition $\phi_t$ and use subscripts to

denote conditioning.)

$$\mathbb{E}_{B_t, \boldsymbol{\zeta}_{j \in [M]}} \big[ \boldsymbol{g}_t \mid \phi_t \big] \quad (93)$$

$$= \frac{1}{bM} \mathbb{E}_{B_t} \left[ \sum_{\boldsymbol{z} \in B_t} \sum_{j=1}^{M} \mathbb{E}_{\boldsymbol{\zeta}_j} \big[ \nabla_\phi \ell_\tau(\phi_t; \boldsymbol{z}, \boldsymbol{\zeta}_j) \big] \right] \quad (94)$$

$$= \nabla_\phi \frac{1}{bM} \mathbb{E}_{B_t} \left[ \sum_{\boldsymbol{z} \in B_t} \sum_{j=1}^{M} \mathbb{E}_{\boldsymbol{\zeta}_j} \big[ \ell_\tau(\phi_t; \boldsymbol{z}, \boldsymbol{\zeta}_j) \big] \right] \quad (95)$$

$$= \nabla_\phi \frac{1}{b} \mathbb{E}_{B_t} \left[ \sum_{\boldsymbol{z} \in B_t} \mathbb{E}_{\boldsymbol{\zeta}} \big[ \ell_\tau(\phi_t; \boldsymbol{z}, \boldsymbol{\zeta}) \big] \right] \quad (96)$$

$$= \nabla_\phi L_{\tau,S}(\phi_t), \quad (97)$$

where the expectation is taken with respect to the randomness of the mini-batch sampling and the auxiliary noise.

Now, we show that the estimator has bound variance. By the law of total variance, conditioning on the mini-batch $B_t$, we have

$$\mathbb{E}\big[ \|\boldsymbol{g}_t - \mathbb{E}\boldsymbol{g}_t\|^2 \big] = \mathbb{E}_{B_t} \Big[ \mathbb{E}\big[ \|\boldsymbol{g}_t - \mathbb{E}[\boldsymbol{g}_t \mid B_t]\|^2 \mid B_t \big] \Big]$$
$$+ \mathbb{E}_{B_t} \Big[ \|\mathbb{E}[\boldsymbol{g}_t \mid B_t] - \mathbb{E}\boldsymbol{g}_t\|^2 \Big]. \quad (98)$$

We first analyze the conditional expectation. For a fixed mini-batch $B_t$, we have

$$\mathbb{E}[\boldsymbol{g}_t \mid B_t] = \frac{1}{bM} \sum_{\boldsymbol{z} \in B_t} \sum_{j=1}^{M} \mathbb{E}_{\boldsymbol{\zeta}_j} \Big[ \nabla_\phi \ell_\tau(\phi_t; \boldsymbol{z}, \boldsymbol{\zeta}_j) \Big] \quad (99)$$

$$= \frac{1}{b} \sum_{\boldsymbol{z} \in B_t} \mathbb{E}_{\boldsymbol{\zeta}} \big[ \bar{\boldsymbol{g}}_M(\phi_t; \boldsymbol{z}) \big] \quad (100)$$

$$= \frac{1}{b} \sum_{\boldsymbol{z} \in B_t} m(\phi_t; \boldsymbol{z}). \quad (101)$$

where

$$\bar{\boldsymbol{g}}_M(\phi; \boldsymbol{z}) = \frac{1}{M} \sum_{j=1}^{M} \nabla_\phi \ell_\tau(\phi; \boldsymbol{z}, \boldsymbol{\zeta}_j) \quad (102)$$

denotes the $M$ average sample of auxiliary noise. Taking expectation over $B_t$ yields

$$\mathbb{E}\boldsymbol{g}_t = \mathbb{E}_{B_t} \Big[ \frac{1}{b} \sum_{\boldsymbol{z} \in B_t} m(\phi_t; \boldsymbol{z}) \Big] \quad (103)$$

$$= \frac{1}{N} \sum_{i=1}^{N} m(\phi_t, \boldsymbol{z}_i) \quad (104)$$

$$= \nabla L_{\tau,S}(\phi_t), \quad (105)$$

where the last equality follows from the definition of $L_{\tau,S}$. We now bound the two terms in (98). For the first term,

conditioning on $B_t$,

$$\boldsymbol{g}_t - \mathbb{E}[\boldsymbol{g}_t \mid B_t] = \frac{1}{b} \sum_{\boldsymbol{z} \in B_t} \left( \bar{\boldsymbol{g}}_M(\boldsymbol{\phi}_t; \boldsymbol{z}) - m(\boldsymbol{\phi}_t; \boldsymbol{z}) \right). \tag{106}$$

Using independence of the auxiliary noise variables, we obtain

$$\mathbb{E}\left[ \|\boldsymbol{g}_t - \mathbb{E}[\boldsymbol{g}_t \mid B_t]\|^2 \mid B_t \right] \tag{107}$$

$$= \frac{1}{b^2} \mathbb{E}\left[ \left\| \sum_{\boldsymbol{z} \in B_t} \left( \bar{\boldsymbol{g}}_M(\boldsymbol{\phi}_t; \boldsymbol{z}) - m(\boldsymbol{\phi}_t; \boldsymbol{z}) \right) \right\|^2 \mid B_t \right] \tag{108}$$

$$= \frac{1}{b^2} \sum_{\boldsymbol{z} \in B_t} \mathbb{E}\left[ \|\bar{\boldsymbol{g}}_M(\boldsymbol{\phi}_t; \boldsymbol{z}) - m(\boldsymbol{\phi}_t; \boldsymbol{z})\|^2 \mid B_t \right] \tag{109}$$

$$\leq \frac{\sigma_\zeta^2}{bM} \tag{110}$$

For the second term in (98), using Assumption (A2) and standard properties of mini-batch sampling, we have

$$\mathbb{E}_{B_t}\left[ \|\mathbb{E}[\boldsymbol{g}_t \mid B_t] - \mathbb{E}\boldsymbol{g}_t\|^2 \right] \tag{111}$$

$$= \mathbb{E}_{B_t} \left\| \frac{1}{b} \sum_{\boldsymbol{z} \in B_t} m(\boldsymbol{\phi}_t; \boldsymbol{z}) - \frac{1}{N} \sum_{i=1}^{N} m(\boldsymbol{\phi}_t; \boldsymbol{z}_i) \right\|^2 \tag{112}$$

$$\leq \frac{\sigma_S^2}{b}. \tag{113}$$

Combining the two bounds completes the proof.

**Step 3: Convergence Result** We assume that the empirical objective $L_{\tau,S}(\boldsymbol{\phi})$ is $\beta$-smooth, i.e., $\forall \boldsymbol{\phi}, \boldsymbol{\phi}'$,

$$\|\nabla L_{\tau,S}(\boldsymbol{\phi}) - \nabla L_{\tau,S}(\boldsymbol{\phi}')\| \leq \beta \|\boldsymbol{\phi} - \boldsymbol{\phi}'\|. \tag{114}$$

The parameter is updated by

$$\boldsymbol{\phi}_{t+1} = \boldsymbol{\phi}_t - \eta \, \boldsymbol{g}_t, \tag{115}$$

where $\boldsymbol{g}_t$ is an unbiased stochastic gradient estimator of $\nabla L_{\tau,S}(\boldsymbol{\phi}_t)$. By $\beta$-smoothness of $L_{\tau,S}$, we have

$$L_{\tau,S}(\boldsymbol{\phi}_{t+1}) \leq L_{\tau,S}(\boldsymbol{\phi}_t) + \langle \nabla L_{\tau,S}(\boldsymbol{\phi}_t), \boldsymbol{\phi}_{t+1} - \boldsymbol{\phi}_t \rangle$$
$$+ \frac{\beta}{2} \|\boldsymbol{\phi}_{t+1} - \boldsymbol{\phi}_t\|^2. \tag{116}$$

Substituting the update rule yields

$$L_{\tau,S}(\boldsymbol{\phi}_{t+1}) \leq L_{\tau,S}(\boldsymbol{\phi}_t) - \eta \langle \nabla L_{\tau,S}(\boldsymbol{\phi}_t), \boldsymbol{g}_t \rangle + \frac{\beta \eta^2}{2} \|\boldsymbol{g}_t\|^2. \tag{117}$$

Taking expectation conditioned on $\boldsymbol{\phi}_t$ and using the unbiasedness $\mathbb{E}[\boldsymbol{g}_t \mid \boldsymbol{\phi}_t] = \nabla L_{\tau,S}(\boldsymbol{\phi}_t)$, we obtain

$$\mathbb{E}\left[ L_{\tau,S}(\boldsymbol{\phi}_{t+1}) \mid \boldsymbol{\phi}_t \right] \leq L_{\tau,S}(\boldsymbol{\phi}_t) - \eta \|\nabla L_{\tau,S}(\boldsymbol{\phi}_t)\|^2$$
$$+ \frac{\beta \eta^2}{2} \mathbb{E}\left[ \|\boldsymbol{g}_t\|^2 \mid \boldsymbol{\phi}_t \right]. \tag{118}$$

By the bounded variance results and let $\sigma^2 = \frac{\sigma_\zeta^2}{bM} + \frac{\sigma_S^2}{b}$, we have

$$\mathbb{E}\left[ \|\boldsymbol{g}_t - \nabla L_{\tau,S}(\boldsymbol{\phi}_t)\|^2 \mid \boldsymbol{\phi}_t \right] \leq \sigma^2, \tag{119}$$

hence

$$\mathbb{E}\left[ \|\boldsymbol{g}_t\|^2 \mid \boldsymbol{\phi}_t \right] \leq \|\nabla L_{\tau,S}(\boldsymbol{\phi}_t)\|^2 + \sigma^2. \tag{120}$$

Substituting back yields

$$\mathbb{E}\left[ L_{\tau,S}(\boldsymbol{\phi}_{t+1}) \mid \boldsymbol{\phi}_t \right]$$
$$\leq L_{\tau,S}(\boldsymbol{\phi}_t) - \eta \left( 1 - \frac{\beta \eta}{2} \right) \|\nabla L_{\tau,S}(\boldsymbol{\phi}_t)\|^2 + \frac{\beta \eta^2}{2} \sigma^2. \tag{121}$$

Assuming $\eta \leq 1/\beta$ and that $L_{\tau,S}$ is bounded below by $L_{\tau,S}^\star$, summing over $t = 0, \ldots, T-1$ gives

$$\frac{1}{T} \sum_{t=0}^{T-1} \mathbb{E}\left[ \|\nabla L_{\tau,S}(\boldsymbol{\phi}_t)\|^2 \right]$$
$$\leq \frac{2 \left( L_{\tau,S}(\boldsymbol{\phi}_0) - L_{\tau,S}^\star \right)}{\eta T} + \beta \eta \sigma^2. \tag{122}$$

$$\square$$

# B. Parametric GMM Approximation of the Ideal NPPR Objective

In this section, we clarify the relationship between the ideal non-parametric probabilistic robustness objective and the finite-dimensional GMM-based estimator used in our implementation. The term "non-parametric" refers to the metric-level formulation: NPPR is defined by optimizing over an admissible set of perturbation distributions, without assuming a fixed perturbation family *a priori*. In practice, however, we use a finite-$K$ Gaussian mixture model (GMM) family as a tractable parametric approximation to this ideal objective.

Let $\mathcal{P}_\varepsilon$ denote the admissible perturbation distribution set supported on the perturbation budget, and let

$$\mathcal{G}_{\mathrm{NPPR}}(\boldsymbol{x}, y) \triangleq \inf_{\omega \in \mathcal{P}_\varepsilon} \mathbb{E}_{\boldsymbol{\varepsilon} \sim \omega(\cdot \mid \boldsymbol{x}, y)} \left[ \mathbf{1}_{h(\boldsymbol{x} + \mathcal{U}(\boldsymbol{\varepsilon})) = y} \right] \tag{123}$$

be the ideal NPPR value. Here, the infimum is taken over the full admissible perturbation distribution set. This corresponds to the most conservative probabilistic robustness

value under the admissible distributional uncertainty. In contrast, the practical estimator used in this paper restricts the perturbation distribution to a $K$-component GMM family. Let

$$\mathcal{Q}_K \subseteq \mathcal{P}_\varepsilon \qquad (124)$$

denote the class of admissible $K$-component GMM perturbation distributions, and define

$$\mathcal{G}_K(\boldsymbol{x}, y) \triangleq \inf_{\omega \in \mathcal{Q}_K} \mathbb{E}_{\boldsymbol{\varepsilon} \sim \omega(\cdot | \boldsymbol{x}, y)} \left[ \mathbf{1}_{h(\boldsymbol{x} + \mathcal{U}(\boldsymbol{\varepsilon})) = y} \right]. \qquad (125)$$

Since $\mathcal{Q}_K$ is a restricted subset of $\mathcal{P}_\varepsilon$, we immediately have

$$\mathcal{G}_{\mathrm{NPPR}}(\boldsymbol{x}, y) \leq \mathcal{G}_K(\boldsymbol{x}, y). \qquad (126)$$

Therefore, the finite-GMM estimator is in general less conservative than the fully unrestricted NPPR objective, because it optimizes over a smaller family of perturbation distributions. On the other hand, the GMM family is more flexible than a single fixed reference distribution such as the standard Gaussian perturbation used in standard probabilistic robustness evaluation. Denote the corresponding standard PR value by

$$\mathcal{G}_{\mathrm{PR}}(\boldsymbol{x}, y) \triangleq \mathbb{E}_{\boldsymbol{\varepsilon} \sim \omega_0(\cdot | \boldsymbol{x}, y)} \left[ \mathbf{1}_{h(\boldsymbol{x} + \mathcal{U}(\boldsymbol{\varepsilon})) = y} \right], \qquad (127)$$

where $\omega_0$ is the fixed reference distribution. If $\omega_0 \in \mathcal{Q}_K$, then

$$\mathcal{G}_K(\boldsymbol{x}, y) \leq \mathcal{G}_{\mathrm{PR}}(\boldsymbol{x}, y). \qquad (128)$$

Combining the two inequalities gives

$$\mathcal{G}_{\mathrm{NPPR}}(\boldsymbol{x}, y) \leq \mathcal{G}_K(\boldsymbol{x}, y) \leq \mathcal{G}_{\mathrm{PR}}(\boldsymbol{x}, y). \qquad (129)$$

This shows that the proposed finite-GMM estimator remains more conservative than standard PR, while it can be less conservative than the fully unrestricted NPPR objective due to the parametric restriction. We next quantify the approximation gap induced by restricting the admissible perturbation family to $\mathcal{Q}_K$. Define

$$\eta_K \triangleq \sup_{\omega \in \mathcal{P}_\varepsilon} \inf_{\widetilde{\omega} \in \mathcal{Q}_K} d_{\mathrm{TV}}(\omega, \widetilde{\omega}), \qquad (130)$$

where $d_{\mathrm{TV}}$ denotes the total variation distance. Since the indicator loss is bounded in $[0, 1]$, for any two perturbation distributions $\omega$ and $\widetilde{\omega}$ we have

$$\left| \mathbb{E}_\omega \left[ \mathbf{1}_{h(\boldsymbol{x} + \mathcal{U}(\boldsymbol{\varepsilon})) = y} \right] - \mathbb{E}_{\widetilde{\omega}} \left[ \mathbf{1}_{h(\boldsymbol{x} + \mathcal{U}(\boldsymbol{\varepsilon})) = y} \right] \right| \leq d_{\mathrm{TV}}(\omega, \widetilde{\omega}). \qquad (131)$$

Consequently,

$$0 \leq \mathcal{G}_K(x, y) - \mathcal{G}_{\mathrm{NPPR}}(x, y) \leq \eta_K. \qquad (132)$$

Thus, the conservativeness gap between the finite-GMM estimator and the ideal NPPR objective is controlled by how well the $K$-component GMM family approximates the worst admissible perturbation distribution in total variation

*Table 6.* **Additional training-stage analysis on a small CIFAR-10 subset using ResNet18.** Larger $K$ and longer optimization lead to smaller estimated PR values.

| $K$ | Epoch | Loss | PR |
|---|---|---|---|
| 1 | 100 | 2.03 | 0.50 |
| 1 | 300 | 1.60 | 0.39 |
| 1 | 500 | 1.42 | 0.34 |
| 3 | 100 | 1.60 | 0.39 |
| 3 | 300 | 1.26 | 0.31 |
| 3 | 500 | 1.08 | 0.25 |

distance. In practice, the choice of $K$ controls the trade-off between expressiveness and computational tractability. A larger $K$ yields a richer perturbation family and can better approximate complex perturbation distributions, while a smaller $K$ gives a more efficient but potentially less conservative estimator. From an application perspective, the $K$ mixture components can also be interpreted as capturing several common perturbation modes, such as fog, motion blur, brightness change, or low-light noise.

We provide additional empirical evidence supporting the interpretation that a richer perturbation family leads to a more conservative robustness estimate. As discussed in Appendix B, the finite-$K$ GMM estimator is a tractable parametric approximation to the ideal NPPR objective, and increasing $K$ enlarges the family of admissible perturbation distributions. Therefore, a larger $K$ is expected to identify more challenging perturbation distributions and produce smaller robustness values.

This trend is consistent with Tab. 2 in the main paper. Under the joint-dependence setting, the robustness value decreases from 78.59 at $K = 3$ to 76.27 at $K = 7$, and further to 73.11 at $K = 12$. We further conduct a training-stage analysis on a small CIFAR-10 subset using ResNet18 with maximum batch size 20. As shown in Tab. 6, the estimated PR value decreases with longer optimization for both $K = 1$ and $K = 3$. Moreover, the $K = 3$ estimator consistently gives smaller robustness values than $K = 1$, suggesting that the richer mixture family yields a more conservative estimate. These results empirically support the role of $K$ as a trade-off between expressiveness and computational tractability.

## C. Details on Bicubic Up-sampling

Since input images typically reside in high-dimensional spaces, the covariance matrix of the perturbation distribution becomes prohibitively large, scaling as $\mathcal{O}(d^2)$ with respect to the input dimension $d$. This quadratic growth renders both storage and computation infeasible when the input dimension is large, as in modern image datasets. To mitigate

*Table 7.* **Configuration summary for NPPR estimation.**

| Setting | Value |
|---|---|
| **Model / Architecture** | |
| Up-sampler | bicubic interpolation |
| norm | $\ell_\infty$ |
| $\epsilon$ | $\{4/255, 8/255, 16/255\}$ |
| **GMM Parameters** | |
| Initialization | uniform |
| Modes $K$ | $\{3, 7, 12\}$ |
| Latent dim. | $\{128, 256\}$ |
| Covariance type | full |
| Hidden dim. | $\{256, 512\}$ |
| Label emb. dim. | $\{64, 128\}$ |
| Label emb. norm. | TRUE |
| **Training Hyperparameters** | |
| Epochs | 50 |
| Learning rate | $\{5 \times 10^{-4}, 2 \times 10^{-2}\}$ |
| LR warmup epochs | 20 |
| LR min | $2 \times 10^{-6}$ |
| Loss type | C&W |
| $\kappa$ | 1 |
| Samples per input | 32 |
| **Annealing Schedule** | |
| $T_\pi$ (init $\to$ final) | $3.0 \to 1.0$ |
| $T_\mu$ (init $\to$ final) | $3.0 \to 1.0$ |
| $T_\sigma$ (init $\to$ final) | $1.5 \to 1.0$ |
| $T_{\text{shared}}$ (init $\to$ final) | $1.5 \to 1.0$ |
| Gumbel anneal | TRUE |
| Gumbel temp (init $\to$ final) | $1.0 \to 0.1$ |

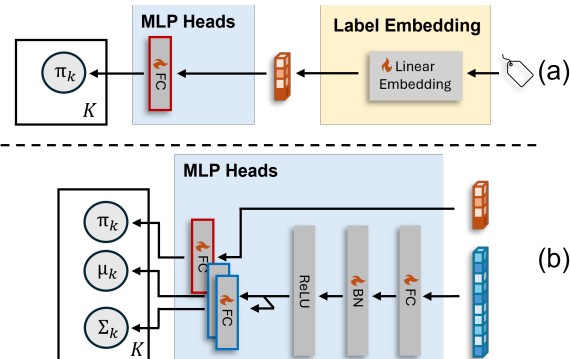

*Figure 4.* **Different dependency constructions.** We employ distinct MLP heads to model different dependency structures. **Panel (a)** illustrates the setting in which the perturbation distribution is conditioned solely on the ground-truth label, whereas **panel (b)** depicts the joint dependency case, where perturbations are conditioned on both the input features and labels, with the labels influencing only the mixture proportions. The label embedding in panel (b) is omitted for clarity, as it is identical to that in panel (a).

this issue, we perform the perturbation modeling in a lower-dimensional space, reducing computational overhead, and subsequently map the perturbations back to the input space using *bicubic interpolation*. This approach is computationally efficient while preserving the spatial smoothness of the perturbations, which has been studied in the robustness-related literature (Dong et al., 2014; Wang et al., 2021b).

Our bicubic up-sampling is composed of a linear mapping and a bicubic interpolation module. For a given pixel position $(x, y)$ in the upsampled image, bicubic interpolation estimates its intensity as a weighted sum of the $4 \times 4$ neighboring pixels in the original image:

$$I'(x, y) = \sum_{m=-1}^{2} \sum_{n=-1}^{2} w(m, x)\, w(n, y)\, I(i + m, j + n), \tag{133}$$

where $I(i + m, j + n)$ denotes the neighboring pixel values and $w(\cdot, \cdot)$ represents the interpolation weights deter-

mined by a cubic convolution kernel. The one-dimensional cubic kernel $w(a)$ is defined as a piecewise cubic polynomial (Keys, 2003):

$$w(a) = \begin{cases} (1.5)|a|^3 - 2.5|a|^2 + 1, & \text{if } |a| < 1, \\ -0.5|a|^3 + 2.5|a|^2 - 4|a| + 2, & \text{if } 1 \le |a| < 2, \\ 0, & \text{otherwise.} \end{cases} \tag{134}$$

This kernel ensures smoothness and locality, producing continuous first derivatives while limiting interpolation to the $4 \times 4$ neighborhood around $(i, j)$. Bicubic up-sampling has been widely adopted as a baseline in image super-resolution and restoration tasks (Dong et al., 2016).

To ensure that the support of the perturbation distribution lies within the prescribed $L_p$-norm ball, we apply the mapping $g_{\mathcal{B}}$, defined as

$$g_{\mathcal{B}} = \gamma \tanh(\cdot), \tag{135}$$

which is a commonly used constraint mechanism in robustness literature (Chen et al., 2022; Huang & Zhang, 2020).

## D. Detailed Experiment Settings

We provide the detailed experimental configurations in Tab. 7. For the independent–perturbation setting with a fixed up-sampler, we adopt a different training strategy from the other cases because this setting is substantially harder to optimize. Specifically, we use a larger learning rate of $2 \times 10^{-2}$ with a cosine cyclical scheduler and a 20-epoch warm-up, which yields the most stable training trajectory.

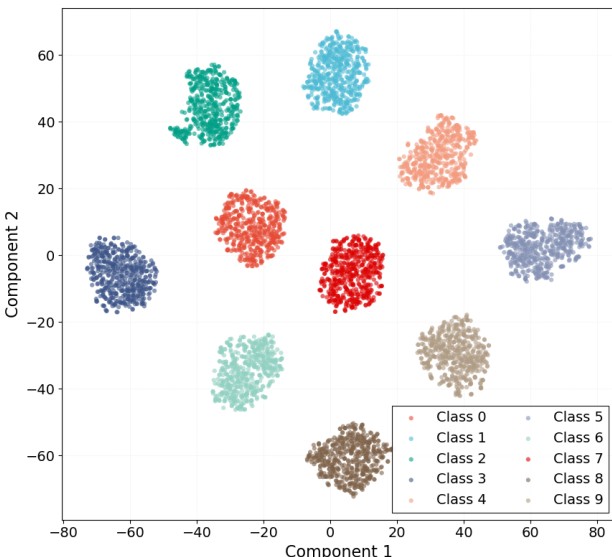

*Figure 5.* **The t-SNE plot for the jointly dependent case.** We additionally visualize the jointly conditioned model for ResNet18 on CIFAR-10. With the added dependence on inputs, the perturbation distributions for different classes become fully disentangled.

For all other dependency settings, including those with a learnable up-sampler, we use a fixed learning rate of $5 \times 10^{-4}$. Except for the runs shown in Fig. 7, which are trained for 200 epochs for visualization purposes, all reported results are trained for 50 epochs.

**Annealing Schedule** We adopt an annealing strategy for the mixture weights as well as the parameters of each mixture-component distribution. This is motivated by the substantial imbalance in the number of parameters associated with the mixture weights, means, and covariance matrices. For example, when using $K = 3$, a latent dimension of 128, and a hidden dimension of 256, the mixture weights require only $3 \times 256$ parameters, whereas the mean and covariance heads require $128 \times 256$ and $128^2 \times 256$ parameters, respectively. Such a disparity can cause optimization to be dominated by the larger parameter groups, leading to suboptimal local minima. To mitigate this imbalance, we apply annealing to stabilize training and prevent premature convergence to poor solutions.

### D.1. Gumbel softmax trick

Training a Gaussian Mixture Model (GMM) within a gradient-based framework requires differentiating through the discrete mixture-selection variable. Specifically, for each perturbation sample $\varepsilon_i$, a categorical latent variable $z_i \in \{1, \ldots, K\}$ determines which Gaussian component generates the sample. Directly sampling $z_i \sim \mathrm{Cat}(\pi_1, \ldots, \pi_K)$ is non-differentiable, preventing backpropagation. To overcome this limitation, we adopt the Gumbel–Softmax (also

known as the Concrete) relaxation (Jang et al., 2017; Maddison et al., 2017), which provides a differentiable approximation to categorical sampling.

The trick relies on the Gumbel perturbation property: if $g_k$ are i.i.d. samples from $\mathrm{Gumbel}(0, 1)$, then

$$z = \arg \max_k \left( \log \pi_k + g_k \right) \qquad (136)$$

is exactly distributed as a categorical random variable with probabilities $\{\pi_k\}_{k=1}^K$. Instead of taking the non-differentiable $\arg \max$, Gumbel–Softmax introduces a temperature-controlled softmax relaxation:

$$\tilde{z}_k = \frac{\exp \left( \left( \log \pi_k + g_k \right) / \tau \right)}{\sum_{j=1}^K \exp \left( \left( \log \pi_j + g_j \right) / \tau \right)}, \qquad k = 1, \ldots, K, \qquad (137)$$

where $\tau > 0$ is a temperature parameter. When $\tau \to 0$, the distribution becomes increasingly "one-hot," recovering a true categorical sample; when $\tau$ is larger, the distribution is smoother, enabling stable gradients.

The reparameterized mixture selection is therefore given by the continuous vector

$$\tilde{\boldsymbol{z}} = (\tilde{z}_1, \ldots, \tilde{z}_K), \qquad (138)$$

which lies in the probability simplex and is fully differentiable with respect to the mixture weights $\pi_k$. This relaxed one-hot vector replaces the discrete indicator and allows the GMM sample to be expressed as:

$$\varepsilon_i = \sum_{k=1}^K \tilde{z}_k \, \mu_k + \sum_{k=1}^K \tilde{z}_k \, \Sigma_k^{1/2} \boldsymbol{\xi}_k, \qquad (139)$$

where $\boldsymbol{\xi}_k \sim \mathcal{N}(0, I)$ is an auxiliary noise variable. Because all operations are differentiable, the entire perturbation generation process is trainable via standard backpropagation.

During training, we anneal the temperature $\tau$ from a higher initial value to a smaller final value, which encourages exploration early on and progressively sharpens the mixture assignments. This annealing strategy stabilizes optimization and prevents premature distribution collapse.

### D.2. Entropy Ratio

Entropy Ratio (ER) quantifies the degree of mode dominance in a Gaussian Mixture Model (GMM). It measures how evenly the mixture weights $\boldsymbol{\pi} = (\pi_1, \ldots, \pi_K)$ are distributed across the $K$ components. Lower ER values indicate that the probability mass is concentrated on a single dominant mode (i.e., mode collapse), whereas values closer to 1 suggest a more uniform mixture distribution.

Formally, ER is defined as

$$\mathrm{ER}(\boldsymbol{\pi}) = \frac{H(\boldsymbol{\pi})}{\log K} = \frac{-\sum_{k=1}^K \pi_k \log \pi_k}{\log K}, \qquad (140)$$

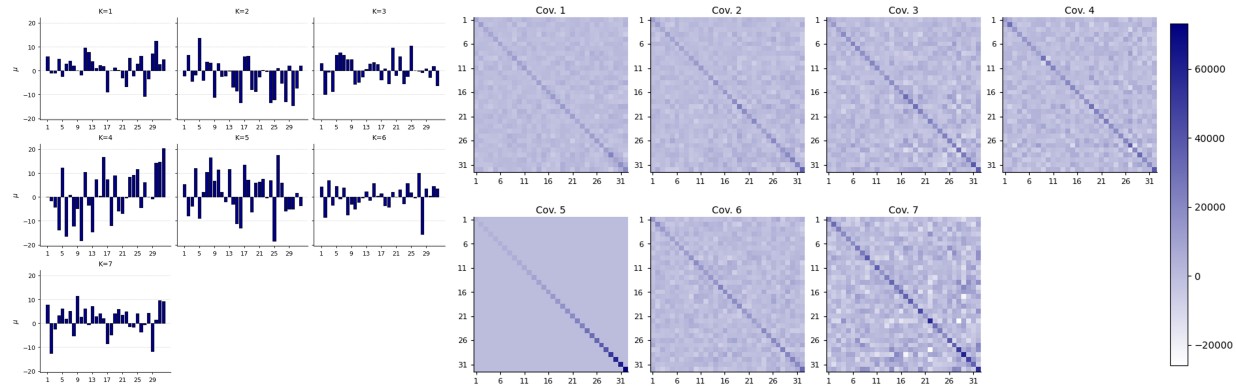

*Figure 6.* **Bar plot and heatmap of mixture component means and covariances.** For the input-dependent case on ResNet18 (CIFAR-10), we randomly select one input and visualize the GMM parameters after reducing the feature dimension to 32 using PCA. Specifically, we display a bar plot of the mixture means and a heatmap of the corresponding covariance matrices.

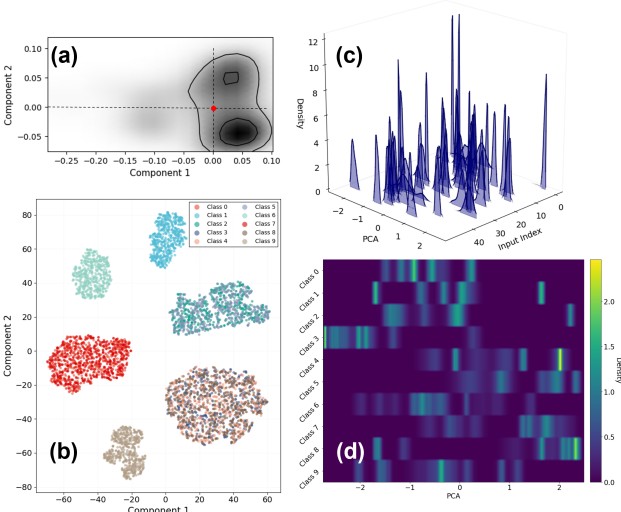

*Figure 7.* **Distribution under different dependence structures** Panel **(a)** shows the PCA-projected contour of the perturbation distribution for the independent case. Panel **(b)** visualizes the t-SNE embeddings of perturbations for the label-dependent case. Panel **(c)** presents PCA-based density plots for 50 randomly selected inputs under the input-dependent setting. Panel **(d)** displays a class-wise heatmap of perturbation densities for the jointly dependent case.

where $H(\boldsymbol{\pi})$ denotes the Shannon entropy of the mixture weights, and $\log K$ is the maximum possible entropy for a $K$-component mixture. This normalization ensures that $\mathrm{ER} \in [0, 1]$, allowing consistent comparison across different values of $K$.

## E. Additional Experiments

Here, we provide additional results that further illustrate the characteristics of the learned perturbation distributions.

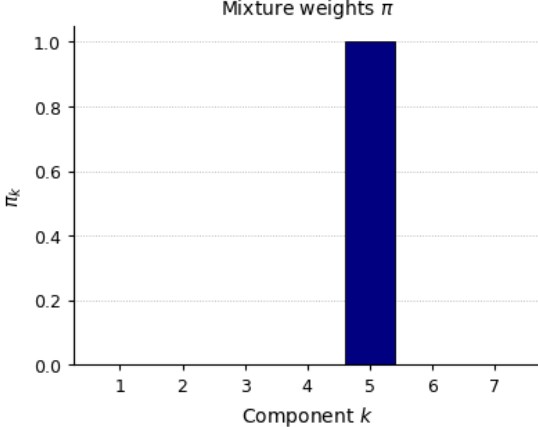

*Figure 8.* **Bar plot of mixture proportions.** We visualize the mixture proportions for the same input and model used in Fig. 6 by plotting the corresponding bar chart.

Fig. 7 illustrates the resulting distributions. Panel (a) presents the perturbation distribution for the independent case, projected onto the first two principal components using PCA. The results show that the learned distribution allocates a greater portion of its probability mass away from the center (marked by the red dot), indicating increased diversity that encourages exploration of decision boundaries near the edges.

Panel (b) visualizes the label-dependent perturbation distribution using t-SNE. The result reveals an interesting structural pattern. Although the distribution is conditioned on the ground-truth labels (10 in total), only 6 distinct clusters are formed. Some classes, such as class 7, are well separated, whereas others, e.g., classes 3 and 4, exhibit strong overlap, indicating shared perturbation characteristics. Fig. 5 in Appendix D presents the t-SNE visualization for the jointly dependent case, where samples from all classes are clearly

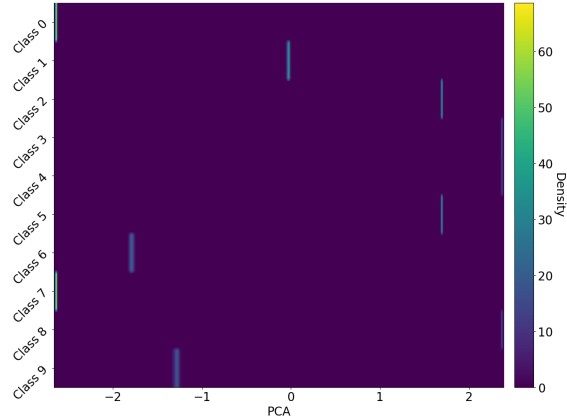

*Figure 9.* **Class-wise heatmap of perturbation densities for the label-dependent case.** The heatmap is generated from the learned distribution of ResNet on CIFAR-10, using the same experimental setup as panel (d) in Fig. 7.

disentangled.

Panel (c) randomly samples 50 CIFAR-10 inputs and applies PCA to their perturbations to visualize the dominant variation directions. Although an individual input may still exhibit mode collapse, which is consistent with the mixture-weight bar plots in Fig. 8 (Appendix E), different inputs activate distinct perturbation patterns. This diversity in perturbations results in substantially lower NPPR values.

Panel (d) shows the heatmap for the jointly dependent case, where each band represents the perturbation distribution over 100 randomly selected inputs within a given class. Compared to the label-dependent case, the jointly dependent formulation offers greater flexibility in the distribution of each class (cf. Fig. 9 in Appendix E).

Fig. 5 presents the t-SNE visualization corresponding to Fig. 7 panel (b), but using the joint-dependence structure instead of label dependence. As shown, perturbation samples from all classes become clearly disentangled, indicating that joint dependence yields a substantially more diverse and well-separated perturbation distribution than label dependence.

Fig. 6 and 8 present bar plots of the mixture means after applying PCA (reduced to 32 dimensions), heatmaps of the covariance matrices, and bar plots of the mixture proportions. The results show that only a single mixture component remains active, thereby dominating the distribution. The covariance heatmaps further reveal that this dominant component exhibits strong diagonal values with minimal off-diagonal structure, indicating limited correlation across dimensions.

Fig. 9 shows the heatmap of perturbation distributions grouped by labels. Compared with Fig. 7 panel (d), the joint-dependent case exhibits substantially greater diversity

within each label group, indicating a more varied perturbation distribution.

In Tab. 11, NPPR captures the vulnerability trend under natural corruptions more faithfully than fixed-distribution PR baselines. In particular, fixed Gaussian, Uniform, and Laplace PR often produce overly high robustness scores, especially for adversarially trained models, making them less informative about actual model fragility. In contrast, even though our feature extractor is trained on standard rather than adversarially trained models, NPPR still reveals model vulnerability and aligns better with corruption robustness, suggesting that the learned non-parametric distribution provides a more sensitive robustness estimate when fixed perturbation distributions largely fail.

Tab. 12 shows the extended experiments of our proposed pipeline using ResNet18 on CIFAR-10. As indicated, the results match our observations in the ablation study in Tab. 2. In addition, we include a setting where we directly optimize in the input space without the up-sampling module. Due to the large input dimensionality, this greatly increases the number of parameters and computational cost, and leads to a very unstable training trajectory. Without an up-sampler, it also hardly generalizes to high-resolution images, e.g., $3 \times 224 \times 224$ in ImageNet. Hence, we exclude it from our main experiments, though it has a better performance. In the table, in addition to the estimated NPPR on the test set, $\widehat{\mathcal{G}}_{\text{NPPR}_{\text{test}}}$, and the entropy ratio, we also report the maximum, minimum, and standard deviation of the mixture proportions $\pi$.

### E.1. Additional Results on Scalability, Inference Cost, and Training Overhead

In this section, we provide additional computational cost analysis for NPPR from both the inference and training perspectives. Since NPPR estimates robustness under an unknown perturbation distribution by fitting a conservative perturbation model, it introduces extra computational cost compared with standard PR methods that assume a fixed perturbation family. To give a more complete picture of this overhead, we report both inference wall-clock time and estimator training time across different architectures and datasets.

**Inference wall-clock time** Tab. 8 summarizes the wall-clock inference time across all considered architectures and datasets, including standard PR under several fixed perturbation families, the GMM-based NPPR estimator, and attack-based baselines. Overall, NPPR is consistently only moderately slower than standard PR variants such as Gaussian, uniform, and Laplace PR, while remaining substantially faster than stronger attack-based baselines such as AutoAttack (AA).

*Table 8.* **Inference wall-clock time across datasets and models (seconds).** NPPR corresponds to the GMM-based estimator. Compared with fixed-family PR baselines, NPPR introduces only a modest additional inference cost, while remaining substantially faster than AutoAttack. Values in parentheses denote the per-epoch training wall-clock time (seconds) of NPPR on a single RTX 3090.

| Dataset | Model | $\widehat{\mathcal{G}}_{\text{NPPR}}$ | $\widehat{\mathcal{G}}_{\text{PR}_{\text{Gaussian}}}$ | $\widehat{\mathcal{G}}_{\text{PR}_{\text{Uniform}}}$ | $\widehat{\mathcal{G}}_{\text{PR}_{\text{Laplace}}}$ | $\widehat{\mathcal{G}}_{\text{AR}_{\text{PGD}}}$ | $\widehat{\mathcal{G}}_{\text{AR}_{\text{CW}}}$ | $\widehat{\mathcal{G}}_{\text{AR}_{\text{AA}}}$ |
|---|---|---|---|---|---|---|---|---|
| CIFAR-10 | ResNet18 | 3.96(69.45) | 2.55 | 2.55 | 2.63 | 2.96 | 2.70 | 30.20 |
| | ResNet50 | 11.34(154.78) | 9.52 | 9.57 | 9.65 | 8.25 | 7.93 | 78.18 |
| | WRN50 | 16.15(226.24) | 14.07 | 14.07 | 14.16 | 13.76 | 13.40 | 127.12 |
| | VGG16 | 16.31(207.23) | 14.11 | 14.16 | 14.28 | 11.23 | 11.01 | 106.80 |
| CIFAR-100 | ResNet18 | 3.98(64.62) | 2.54 | 2.54 | 2.63 | 3.04 | 2.70 | 96.42 |
| | ResNet50 | 11.34(154.41) | 9.50 | 9.56 | 9.66 | 8.21 | 7.83 | 79.04 |
| | WRN50 | 16.34(229.04) | 14.21 | 14.22 | 14.31 | 13.78 | 13.45 | 120.93 |
| | VGG16 | 16.49(195.08) | 14.56 | 14.60 | 14.81 | 11.40 | 11.17 | 96.23 |
| TinyImageNet | ResNet18 | 8.36(203.23) | 6.57 | 6.59 | 6.93 | 6.11 | 5.78 | 40.08 |
| | ResNet50 | 23.60(565.14) | 20.98 | 21.03 | 21.39 | 18.37 | 18.04 | 143.27 |
| | WRN50 | 37.16(907.58) | 34.29 | 34.26 | 34.56 | 30.28 | 29.66 | 244.73 |
| | VGG16 | 43.73(994.21) | 40.41 | 40.56 | 41.03 | 28.66 | 28.10 | 225.54 |

*Table 9.* **NPPR across evaluation radii and architectures (%).** Estimators are trained at an $L_\infty$ perturbation radius of $16/255$ and evaluated under different $L_\infty$ radii across multiple datasets and architectures. All values are percentages.

| Dataset | Model | 4/255 | 8/255 | 16/255 | 32/255 |
|---|---|---|---|---|---|
| CIFAR-10 | ResNet18 | 96.75 | 90.99 | 76.27 | 52.49 |
| | ResNet50 | 98.51 | 94.99 | 86.81 | 68.41 |
| | WRN50 | 97.76 | 93.15 | 81.32 | 59.45 |
| | VGG16 | 96.27 | 90.06 | 74.47 | 49.72 |
| CIFAR-100 | ResNet18 | 95.71 | 84.10 | 58.60 | 27.74 |
| | ResNet50 | 96.03 | 84.46 | 58.92 | 27.49 |
| | WRN50 | 94.06 | 82.19 | 58.83 | 28.26 |
| | VGG16 | 94.26 | 79.92 | 52.20 | 23.18 |
| TinyImageNet | ResNet18 | 93.24 | 78.43 | 53.29 | 27.09 |
| | ResNet50 | 95.37 | 85.10 | 64.37 | 31.75 |
| | WRN50 | 94.83 | 83.40 | 59.86 | 28.89 |
| | VGG16 | 94.94 | 82.64 | 59.00 | 25.51 |

For example, on ResNet18 with CIFAR-10, NPPR requires 3.96 seconds, compared with 2.55 seconds for PR-Gaussian and 30.20 seconds for AA. On a larger configuration, namely WRN50 with TinyImageNet, NPPR requires 37.16 seconds, whereas AA requires 244.73 seconds. These results suggest that incorporating perturbation-distribution learning leads to a moderate increase in inference cost relative to fixed-family PR baselines, but still preserves a clear efficiency advantage over strong attack-based evaluations.

**Training wall-clock time** In addition to inference cost, we also report, in parentheses in the NPPR column, the per-epoch training wall-clock time of the NPPR estimator. This is included to provide a compact view of both inference and estimator-fitting cost in a single table. NPPR incurs extra training overhead because the perturbation estimator

itself must be optimized. To quantify this overhead, we measure the wall-clock training time of the GMM-based NPPR estimator on a single RTX 3090 and report the per-epoch time.

As expected, the fitting cost grows with both dataset scale and model complexity. For example, on CIFAR-10, the per-epoch training time increases from 69.45 seconds for ResNet18 to 226.24 seconds for WRN50. On TinyImageNet, the corresponding cost further increases to 203.23 seconds for ResNet18, 565.14 seconds for ResNet50, 907.58 seconds for WRN50, and 994.21 seconds for VGG16. This indicates that, although NPPR remains practical at the scales considered in this work, estimator fitting constitutes the dominant source of additional computational overhead.

At the same time, these results suggest several possible directions for improving efficiency, such as adopting a lighter perturbation estimator, sharing feature extractors, or reducing the frequency of estimator updates during training.

### E.2. Cross-radius behavior across architectures and datasets.

To test whether the above trend is specific to a single architecture–dataset pair, we additionally fix the training radius at $16/255$ and evaluate the learned estimator across multiple models and datasets. The results are reported in Tab. 9. The same monotone trend holds consistently across all tested settings: larger evaluation radii lead to lower NPPR estimates. This behavior is observed for all architectures and datasets, including CIFAR-10, CIFAR-100, and Tiny-ImageNet.

For example, on ResNet50/CIFAR-10, the NPPR score

*Table 10.* **Comparison of NPPR with a weak-adversary baseline and intermediate-stress baselines on ResNet18.** The weak-adversary baseline uses one-step PGD with random Gaussian initialization of variance 0.1. The intermediate-stress formulation follows Rice et al. (Rice et al., 2021) and is normalized by the standard loss.

| Dataset | $\widehat{\mathcal{G}}_{\text{NPPR}}$ | $\widehat{\mathcal{G}}_{\text{AR}_{\text{PGD-1}}}$ | $q = 1/\text{std}$ | $q = 10/\text{std}$ | $q = 100/\text{std}$ | $q = 1000/\text{std}$ |
|---|---|---|---|---|---|---|
| CIFAR-10 | 0.76 | 0.87 | 0.18 | 0.54 | 0.86 | 0.91 |
| CIFAR-100 | 0.59 | 0.72 | 0.36 | 0.62 | 0.92 | 0.98 |
| TinyImageNet | 0.53 | 0.77 | 0.63 | 0.75 | 0.98 | 1.04 |

decreases from 98.51 at 4/255 to 94.99 at 8/255, 86.81 at 16/255, and 68.41 at 32/255. Similarly, on VGG16/TinyImageNet, it decreases from 94.94 to 82.64, 59.00, and 25.51. Overall, these results suggest that the learned NPPR estimator remains stable under train/test radius mismatch, while still reflecting the expected degradation in robustness as the perturbation radius increases.

## E.3. Comparison with Weak-Adversary and Intermediate-Stress Baselines

We further compare NPPR with two related baselines on ResNet18, i.e., a weak-adversary baseline based on one-step PGD with random Gaussian initialization (variance 0.1), and the intermediate-stress formulation of Rice et al. (Rice et al., 2021), where the latter is normalized by the standard loss for easier comparison. The goal of this comparison is to examine whether NPPR can be reduced to a weak attack score or simply interpreted as another tuned intermediate-severity robustness metric.

Tab. 10 shows that NPPR behaves differently from both baselines across datasets. On CIFAR-10, NPPR is 0.76, compared with 0.87 for the weak-adversary baseline. The intermediate-stress values vary substantially with the stress parameter $q$, ranging from 0.18 to 0.91. A similar pattern is observed on CIFAR-100 and TinyImageNet. In particular, for larger $q$, the intermediate-stress score approaches or even exceeds 1, while NPPR remains much lower. This indicates that NPPR is not simply another tuned intermediate-severity score.

Conceptually, the distinction is also important. The intermediate-stress formulation of Rice et al. (Rice et al., 2021) studies robustness notions between average-case and worst-case evaluation under a fixed perturbation measure. In contrast, NPPR is motivated by uncertainty over the perturbation distribution itself, and evaluates robustness conservatively over an admissible family of perturbation distributions. Therefore, NPPR addresses a different source of uncertainty and should be viewed as complementary rather than equivalent to these baselines.

*Table 11.* **Robustness evaluation across datasets and models (%).** NPPR refers to the GMM-based estimator. We report clean accuracy, probabilistic robustness (PR), adversarial robustness (AR), and corruption robustness under both standard and adversarial training. **Note.** S&P, MB, Bright., and JPEG denote Salt-and-Pepper noise, Motion Blur, Brightness, and JPEG compression, respectively. VGG16 is excluded from adversarial training due to convergence failure. PGD/CW are evaluated with 3 steps for standardly trained models and 10 steps for adversarially trained models; adversarial training uses PGD10. Corruptions are evaluated at severity level 1, and all robustness results are normalized by the corresponding clean accuracy.

| Dataset | Model | Clean Acc. | $\widehat{\mathcal{G}}_{\mathrm{NPPR}}$ (Ours) | $\widehat{\mathcal{G}}_{\mathrm{PR_{Gaussian}}}$ | $\widehat{\mathcal{G}}_{\mathrm{PR_{Uniform}}}$ | $\widehat{\mathcal{G}}_{\mathrm{PR_{Laplace}}}$ | $\widehat{\mathcal{G}}_{\mathrm{AR_{PGD}}}$ | $\widehat{\mathcal{G}}_{\mathrm{AR_{CW}}}$ | $\widehat{\mathcal{G}}_{\mathrm{AR_{AA}}}$ | S&P | MB | Brightness | JPEG |
|---|---|---|---|---|---|---|---|---|---|---|---|---|---|
| | | | | | **Standard Training** | | | | | | | | |
| CIFAR-10 | ResNet18 | 87.72 | 76.29 | 95.28 | 97.21 | 95.13 | 3.10 | 3.30 | 0.00 | 83.63 | 73.65 | 99.66 | 97.31 |
| | ResNet50 | 90.75 | 86.84 | 92.42 | 94.94 | 92.23 | 5.80 | 6.05 | 0.00 | 88.02 | 76.45 | 100.11 | 95.93 |
| | WRN50 | 91.13 | 81.32 | 94.20 | 96.25 | 94.02 | 6.94 | 7.24 | 0.01 | 88.98 | 77.30 | 100.26 | 96.13 |
| | VGG16 | 92.28 | 74.53 | 89.08 | 93.36 | 88.76 | 0.49 | 0.31 | 0.00 | 84.90 | 76.56 | 99.90 | 93.84 |
| CIFAR-100 | ResNet18 | 63.38 | 58.65 | 86.51 | 91.48 | 86.17 | 2.45 | 3.02 | 0.02 | 69.03 | 60.18 | 99.40 | 92.21 |
| | ResNet50 | 70.57 | 58.93 | 80.73 | 86.78 | 80.31 | 3.47 | 3.69 | 0.01 | 76.39 | 59.33 | 100.18 | 88.51 |
| | WRN50 | 72.02 | 58.88 | 82.88 | 88.24 | 82.58 | 4.56 | 4.94 | 0.01 | 77.20 | 59.50 | 99.89 | 89.03 |
| | VGG16 | 71.02 | 52.13 | 75.86 | 83.48 | 75.44 | 0.98 | 0.81 | 0.00 | 78.26 | 56.62 | 99.94 | 85.24 |
| TinyImageNet | ResNet18 | 56.99 | 53.20 | 92.26 | 95.17 | 92.07 | 0.45 | 0.52 | 0.00 | 71.61 | 33.78 | 99.96 | 99.33 |
| | ResNet50 | 70.39 | 64.46 | 92.09 | 95.19 | 91.90 | 3.90 | 2.12 | 0.00 | 72.79 | 42.36 | 99.74 | 99.76 |
| | WRN50 | 73.74 | 59.88 | 93.60 | 96.19 | 93.43 | 4.79 | 3.18 | 0.00 | 72.31 | 54.85 | 99.76 | 99.65 |
| | VGG16 | 66.95 | 59.01 | 93.07 | 95.70 | 92.87 | 4.12 | 0.21 | 0.00 | 70.71 | 42.30 | 100.16 | 100.10 |
| | | | | | **Adversarial Training (PGD-10)** | | | | | | | | |
| CIFAR-10 | ResNet18 | 72.42 | 95.59 | 99.78 | 99.84 | 99.76 | 60.67 | 56.55 | 51.53 | 95.02 | 89.34 | 98.34 | 99.35 |
| | ResNet50 | 75.29 | 95.66 | 99.66 | 99.80 | 99.65 | 59.74 | 56.74 | 51.27 | 92.92 | 89.27 | 97.84 | 98.75 |
| | WRN50 | 78.83 | 96.33 | 99.86 | 99.97 | 99.87 | 58.33 | 55.80 | 50.35 | 92.83 | 88.16 | 98.20 | 98.95 |
| CIFAR-100 | ResNet18 | 49.39 | 89.54 | 99.96 | 99.95 | 99.92 | 43.88 | 39.89 | 34.64 | 90.61 | 81.80 | 97.67 | 97.89 |
| | ResNet50 | 52.14 | 85.88 | 100.06 | 100.09 | 100.07 | 40.26 | 38.01 | 32.83 | 88.70 | 81.40 | 98.85 | 97.10 |
| | WRN50 | 21.10 | 94.28 | 100.11 | 100.00 | 100.03 | 57.44 | 45.73 | 40.28 | 100.00 | 90.43 | 96.35 | 99.62 |
| TinyImageNet | ResNet18 | 21.34 | 91.06 | 99.70 | 99.78 | 99.73 | 44.05 | 32.71 | 27.65 | 90.58 | 72.91 | 94.89 | 99.39 |
| | ResNet50 | 27.94 | 93.21 | 99.79 | 99.72 | 99.76 | 39.30 | 32.14 | 27.27 | 89.08 | 73.51 | 95.85 | 99.79 |
| | WRN50 | 33.37 | 88.95 | 99.84 | 99.93 | 99.81 | 30.33 | 25.77 | 21.22 | 88.97 | 66.50 | 96.19 | 99.55 |

*Table 12.* **Extended Experimental Results of ResNet18 on CIFAR-10 (%).** We report the GMM-based NPPR results under different perturbation budgets $\gamma \in \{4, 8, 16\}/255$. Since the diagnostic statistics are identical across the three perturbation budgets for each configuration, we report them only once.

| Config. | | $\widehat{\mathcal{G}}_{\text{NPPR}}$ | | | Diagnosis | | | |
|---|---|---|---|---|---|---|---|---|
| $K$ | Decoder | 4/255 | 8/255 | 16/255 | ER($\pi$) | Max($\pi$) | Min($\pi$) | Std.($\pi$) |
| **(I) Independent Perturbations** | | | | | | | | |
| 3 | None | 97.53 | 86.97 | 52.05 | 0.00 | 100.00 | 0.00 | 57.74 |
| 3 | Non-trainable | 97.59 | 91.56 | 74.99 | 0.21 | 99.98 | 0.01 | 57.71 |
| 3 | Trainable | 98.12 | 93.62 | 81.27 | 91.35 | 54.43 | 20.94 | 18.37 |
| 7 | None | 97.11 | 85.52 | 52.73 | 0.00 | 100.00 | 0.00 | 37.80 |
| 7 | Non-trainable | 97.47 | 91.01 | 73.28 | 0.09 | 99.98 | 0.00 | 37.79 |
| 7 | Trainable | 97.51 | 91.10 | 73.25 | 94.37 | 31.49 | 8.38 | 7.93 |
| 12 | None | 97.69 | 87.18 | 55.04 | 0.00 | 100.00 | 0.00 | 28.87 |
| 12 | Non-trainable | 97.50 | 91.14 | 73.53 | 0.08 | 99.98 | 0.00 | 28.86 |
| 12 | Trainable | 97.70 | 91.92 | 74.43 | 92.43 | 20.11 | 3.83 | 5.83 |
| **(II) Input-Dependent Perturbations ($x$-dependent)** | | | | | | | | |
| 3 | None | 98.20 | 92.06 | 68.96 | 70.13 | 72.06 | 9.34 | 33.85 |
| 3 | Non-trainable | 98.43 | 94.11 | 85.36 | 57.00 | 77.45 | 3.22 | 39.05 |
| 3 | Trainable | 98.55 | 95.01 | 84.76 | 69.45 | 72.37 | 8.88 | 34.16 |
| 7 | None | 98.30 | 91.86 | 69.00 | 52.95 | 65.20 | 0.00 | 23.39 |
| 7 | Non-trainable | 98.59 | 94.68 | 82.18 | 47.79 | 69.38 | 0.00 | 25.19 |
| 7 | Trainable | 98.67 | 95.01 | 84.17 | 54.28 | 67.08 | 0.03 | 23.92 |
| 12 | None | 98.13 | 92.71 | 69.65 | 44.72 | 70.84 | 0.00 | 19.93 |
| 12 | Non-trainable | 98.25 | 94.08 | 82.43 | 47.02 | 64.17 | 0.00 | 18.22 |
| 12 | Trainable | 98.46 | 94.43 | 83.31 | 48.74 | 64.85 | 0.00 | 18.36 |
| **(III) Label-Dependent Perturbations ($y$-dependent)** | | | | | | | | |
| 3 | None | 94.88 | 81.82 | 45.86 | 99.02 | 40.31 | 29.18 | 6.08 |
| 3 | Non-trainable | 96.33 | 90.42 | 75.79 | 99.22 | 39.46 | 29.26 | 5.40 |
| 3 | Trainable | 95.48 | 87.62 | 69.08 | 99.14 | 39.87 | 29.62 | 5.68 |
| 7 | None | 93.81 | 78.50 | 36.92 | 73.68 | 39.45 | 0.00 | 14.78 |
| 7 | Non-trainable | 95.43 | 88.56 | 72.05 | 72.81 | 40.14 | 0.01 | 15.20 |
| 7 | Trainable | 95.57 | 86.51 | 64.88 | 87.29 | 30.16 | 0.00 | 9.70 |
| 12 | None | 94.12 | 78.71 | 39.21 | 54.65 | 50.20 | 0.00 | 14.70 |
| 12 | Non-trainable | 96.10 | 90.45 | 75.76 | 70.64 | 20.78 | 0.01 | 9.35 |
| 12 | Trainable | 95.15 | 86.27 | 64.87 | 76.59 | 20.24 | 0.01 | 8.25 |
| **(IV) Joint-Dependent Perturbations ($x, y$-dependent)** | | | | | | | | |
| 3 | None | 96.78 | 87.27 | 58.10 | 93.94 | 50.08 | 20.74 | 15.10 |
| 3 | Non-trainable | 96.66 | 90.90 | 79.20 | 99.23 | 39.49 | 29.65 | 5.37 |
| 3 | Trainable | 97.14 | 92.38 | 78.59 | 99.16 | 39.77 | 29.49 | 5.61 |
| 7 | None | 95.83 | 84.57 | 51.76 | 97.36 | 20.35 | 9.92 | 4.99 |
| 7 | Non-trainable | 96.18 | 89.79 | 73.68 | 87.06 | 29.73 | 0.00 | 9.78 |
| 7 | Trainable | 96.77 | 90.98 | 76.27 | 89.63 | 20.86 | 0.00 | 7.99 |
| 12 | None | 95.59 | 83.10 | 47.73 | 76.09 | 20.43 | 0.00 | 8.29 |
| 12 | Non-trainable | 95.84 | 88.94 | 72.68 | 73.83 | 30.23 | 0.00 | 9.38 |
| 12 | Trainable | 96.07 | 89.29 | 73.11 | 73.31 | 31.52 | 0.00 | 9.62 |

