# OpenReview forum: "Non-Parametric Probabilistic Robustness: A Conservative Risk Estimator under Unknown Perturbation Distributions"
_ICML.cc/2026/Conference — ICML 2026 regular_

### Official Review · Reviewer_EzTm · 2026-02-17

**Soundness:** 3
**Presentation:** 2
**Significance:** 3
**Originality:** 2
**Overall Recommendation:** 4
**Confidence:** 4

**Summary:**

The paper introduces Non-Parametric Probabilistic Robustness (NPPR) to address the limitations of standard Probabilistic Robustness (PR), which relies on unrealistic assumptions of fixed perturbation distributions. The authors propose learning a conservative perturbation distribution via a parameterized Gaussian Mixture Model to estimate the worst-case PR within a prescribed budget, effectively positioning NPPR as a middle ground between PR and Adversarial Robustness (AR). Theoretical analysis confirms that NPPR is mathematically bounded by AR and PR, and experiments demonstrate that the learned distribution yields more conservative robustness estimates compared to fixed baselines like Gaussian noise.

**Compliance With Llm Reviewing Policy:**

Affirmed.

**Final Justification:**

The rebuttal helped clarify the paper’s motivation and empirical positioning, but my main concerns remain only partially resolved. I still do not see clear evidence that NPPR provides distinct diagnostic value beyond simpler intermediate-strength baselines, and I remain concerned that the learned perturbation family may function more like an attack-aligned prior than a genuinely distinct uncertainty model. Overall, I view this paper as borderline, so I keep my original score.

---

**After Reply Rebuttal Comment**

Thank you for the additional clarification. This response makes the intended role of NPPR much clearer to me, and I appreciate the added comparisons to weak-adversary and intermediate-stress baselines as well as the additional experiments on the perturbation-family design. In particular, the latter experiments make the framework look less reducible to an attack-aligned construction and address my main conceptual concern. Although I am still not fully convinced about the precise necessity of NPPR, I will raise my score accordingly and hope these clarifications are reflected clearly in the revised version.

**Key Questions For Authors:**

During rebuttal, I would appreciate it if the authors could first address the weaknesses listed above; the questions below are additional clarification requests.

1.	How should the reported wall time be interpreted, and what is it being compared against? Please clarify whether it includes only NPPR evaluation or also the optimization cost to learn the GMM parameters. It would also help to report runtimes for standard PR estimation and common AR evaluations under the same hardware so the relative efficiency of NPPR is clear.

2.	The paper considers four dependency types for the learned perturbation distribution. Could the authors clarify the motivation for separating these cases, and whether the joint input–label conditioning is intended to be the primary recommended setting?


3.	Could the authors comment on whether NPPR is empirically observed to converge to AR?

**Limitations:**

yes

**Strengths And Weaknesses:**

## Strength


1.	The paper establishes a clear conceptual framework for the robustness spectrum between AR and PR, supported by a logically consistent formal definition of NPPR and theoretical statements that clarify its relationship to existing metrics.

2.	The proposed estimation pipeline is easy to follow, offering a structured approach to learning the perturbation distribution.


## Weakness


1.	The practical interpretability of the NPPR score remains unclear. The learned perturbation distribution does not naturally map to concrete physical phenomena in the way benchmarks like Common Corruptions do, making it hard to understand what operational risk an NPPR value represents. The paper would be stronger if it imposed realistic structure on the learned perturbations and included an application-driven case study demonstrating that NPPR better reflects failure rates under realistic shifts than standard probabilistic baselines.

2.	Empirically, NPPR is mainly shown to lie numerically between PR and AR, but it is not yet clear what additional diagnostic value this provides beyond simpler alternatives. In particular, intermediate robustness levels can often be produced by calibrated weak-adversary evaluations or other standard stress tests tuned to matched severity. Including such baselines and analyzing whether NPPR highlights different error patterns or vulnerability types would clarify whether NPPR offers insights beyond conventional tools.

3.	Appendix D suggests mode collapse, where for many inputs the learned mixture is effectively dominated by a single component. This raises the concern that the learned distribution may often behave like a near single-component Gaussian concentrated around an adversarially salient direction, rather than a genuinely diverse mixture. A direct comparison to a simpler gradient-guided probabilistic baseline, such as Gaussian noise centered around a gradient-based shift, would help justify the added complexity and clarify what the learned distribution contributes.


4.	The paper does not clearly specify how the base classifiers are trained (standard vs. adversarial training, or other training methods: PR-AT?) or the exact AR evaluation settings. As a result, the reported PGD/CW robustness is hard to interpret and appears inconsistent with typical expectations for both natural and AT models. The authors should fully report the classifier training recipe and the attack hyperparameters, and include a standardized baseline AutoAttack.


5.	Regardless of how the base classifiers are trained, the paper would benefit from a systematic evaluation across both standard and adversarially trained models. NPPR is motivated as a complementary assessment notion when worst-case robustness is either too pessimistic or not sufficiently informative, but this claim is best tested by reporting AR, PR, and NPPR side-by-side under both training regimes. Such a comparison would clarify whether NPPR provides consistent model ranking or additional diagnostic signal beyond worst-case baselines when robustness is non-trivial, and whether its conclusions generalize beyond a single training setup.



6.	The paper’s motivation and interpretation appear closely related to prior work [1] on robustness between the average and worst case, which similarly considers a spectrum whose extremes align with random (average-case) and adversarial (worst-case) robustness. Although NPPR may target a different angle by learning a conservative perturbation distribution, the paper neither cites nor discusses this connection. A brief discussion positioning NPPR relative to this line of work would improve clarity, especially by explaining the conceptual difference and why NPPR is needed beyond existing intermediate robustness formulations.


[1] Rice, Leslie, et al. "Robustness between the worst and average case." Advances in Neural Information Processing Systems 34 (2021): 27840-27851.

---

> ### Author Rebuttal · Authors · 2026-03-31
>
> We thank the reviewer for the detailed and constructive feedback.
> ### (W1,W2,W6) On the interpretation, practical value, and positioning of NPPR
> Regarding interpretation of NPPR, NPPR is **not** intended to replace physically grounded benchmarks such as Common Corruptions, nor is it just an arbitrary score between PR and AR. Its purpose is to provide a **conservative probabilistic robustness estimate when the perturbation distribution is unknown**. As a simple illustration, a structured mixture on CIFAR-10/ResNet18 with Gaussian, Laplace, and salt-and-pepper components yields weights $(0.19,0.22,0.59)$ and PR $=0.42$ (detaileds in revised version) . These weights indicate relative sensitivity, in this case, salt-and-pepper contributes most to the conservative estimate.
>
> To better illustrate, we additionally compared NPPR with fixed-distribution PR baselines, AR evaluations, and common corruptions. NPPR tracks the **overall corruption vulnerability trend** across datasets much better than standard PR baselines. A compact example for ResNet18 is below; all values are computed on clean correctly classified samples.
>
> |Dataset|PR-Gauss|NPPR|AA|S\&P|Motion|
> |---|---:|---:|---:|---:|---:|
> |CIFAR-10|0.99|0.76|0.00|0.84|0.74|
> |CIFAR-100|0.97|0.59|0.00|0.69|0.60|
> |Tiny-ImageNet|0.98|0.53|0.00|0.72|0.34|
>
> Here NPPR decreases from CIFAR-10 to CIFAR-100 to Tiny-ImageNet, matching corruption deterioration much more clearly than fixed-distribution PR, while AR metrics are nearly collapsed to zero in many cases.
> We will credit Rice et al. (2021) in our revision as a key related work, noting that while they define intermediate robustness, NPPR specifically targets distributional uncertainty in probabilistic settings.
> ### (W3,Q2) On the perturbation family design and dependency types
> We agree that the low-entropy behavior in the appendix deserves clearer discussion and should be stated as a limitation. However, this does not make NPPR equivalent to a test-time gradient-guided baseline. In our framework, once the perturbation distribution is learned, evaluation uses only sampling from this learned distribution, as in Algorithm 2. No test-time gradients or adversarial search are used. By contrast, centering noise around a gradient-based shift requires attack information at test time and is therefore closer to attack-based evaluation.
>
> The four dependency types represent progressively richer assumptions on the underlying perturbation mechanisms. Independent for global mechanisms (e.g., background noise), label-dependent for class-specific mechanisms, input-dependent for input/-adaptive mechanisms.
> ### (W4,W5) On evaluation protocol, training setup, and robustness across training regimes
> Currently, all base classifiers are **standardly trained** models. We will move full training details to the appendix and clarify in the main text that reported AR is defined as $1-\text{attack success rate}$, and that all robustness-type metrics except clean accuracy are computed only on **clean correctly classified samples**. We adopted this protocol to reduce scale effects from different clean accuracies and make robustness measures more comparable across models.
>
> Regarding adversarial training. A compact preliminary result on PGD-10 adversarially trained ResNet18 is below (full experiments in revised version).
>
> |Dataset|PR-Gauss|NPPR|AA|
> |---|---:|---:|---:|
> |CIFAR-10|1.00|0.96|0.52|
> |CIFAR-100|1.00|0.90|0.35|
> |Tiny-ImageNet|1.00|0.91|0.28|
>
> These preliminary results suggest that NPPR remains meaningfully separated from both fixed-distribution PR and worst-case AR even when robustness is non-trivial.
> ### (Q1) On computational cost and wall-time interpretation
> The Time (s) reported in Table 1 reflects **only the test-time NPPR evaluation cost**, i.e., Algorithm 2. It does **not** include the offline optimization cost of learning the GMM parameters. In the revision, we will clearly separate them. Timing results show that NPPR is only moderately slower than standard PR.
>
> |Arch/Dataset|PR-Gauss|NPPR|AA|
> |---|---:|---:|---:|
> |ResNet18/CIFAR-10|2.55|3.96|30.20|
> |ResNet50/Tiny-ImageNet|20.98|23.60|143.27|
> |WRN50-2/Tiny-ImageNet|34.29|37.16|244.73|
> ### (Q3) On whether NPPR empirically approaches AR
> We conducted an additional experiment on ResNet18/CIFAR-10 by training the NPPR for 500 epochs on a small subset (20 batches), so that the convergence trend is clearer. On these training batches, the learned NPPR value indeed decreases to a much smaller level. Concretely, for $K=1$, PR decreases from $0.4960$ (epoch 100) to $0.3357$ (epoch 500), and for $K=3$, from $0.3892$ to $0.2530$, with the surrogate loss decreasing consistently as well. This suggests that with longer optimization and a richer perturbation family, the learned conservative distribution becomes increasingly adversarial, qualitatively consistent with the theory that NPPR approaches AR in the expressive limit. A detailed table will be included in the appendix.

---

> > ### Author Rebuttal · Reviewer_EzTm · 2026-04-03
> >
> > Thank you for the detailed rebuttal and the additional experiments. I appreciate the authors’ efforts to clarify the positioning of NPPR and to address my concerns.
> >
> > **Regarding the practical value**
> >
> > I agree that the additional corruption-trend comparison helps show that NPPR is not merely an arbitrary number between PR and AR. However, my main concern remains unresolved: these results still do not explain why NPPR is needed beyond simpler and more intuitive intermediate-strength baselines. Showing that NPPR correlates with common corruption trends, or lies numerically between PR-Gauss and AA, is not yet evidence of distinct diagnostic value. What remains unclear is what unique insight NPPR provides that simpler calibrated weak-adversary or attack-inspired stress-test baselines would miss, and therefore why NPPR should be regarded as a distinct robustness notion rather than one particular way of constructing a conservative intermediate stress distribution.
> >
> > **Regarding the perturbation family design**
> >
> > I understand that NPPR does not use adversarial search at test time, but my conceptual concern remains: the issue is not the test-time procedure itself, but that the perturbation distribution is explicitly learned through offline white-box optimization against the frozen classifier and then reused at evaluation time through sampling. Given the low-entropy behavior acknowledged in the rebuttal, my concern is that the learned GMM may function less as a genuinely diverse uncertainty model and more as an amortized, attack-aligned stochastic prior. In that sense, the distinction from attack-based evaluation still feels largely procedural rather than conceptual.

---

> > > ### Author Response · Authors · 2026-04-08
> > >
> > > ## Regarding the practical value
> > >
> > > We thank the reviewer for the helpful clarification. We agree that the key issue is not whether NPPR numerically lies between PR and AR, but what **unique insight** it provides.
> > >
> > > That insight comes from defining NPPR over an **admissible perturbation-distribution set**. In the paper, NPPR is the infimum of PR over $\mathcal{P}_\epsilon$, the set of all perturbation distributions supported within the perturbation budget. This corresponds to a minimal-assumption setting in which the budget is known, but the true test-time perturbation law is unknown. In other words, rather than modeling one specific physical corruption or serving as another heuristic intermediate score, it quantifies the most conservative probabilistic robustness level consistent with the available knowledge about perturbations. The GMM is used as a flexible estimator of uncertainty over perturbation distributions, rather than to claim that the true perturbation law is Gaussian-mixture distributed. We agree this was not sufficiently clear in the current draft, and we will revise the paper to make it explicit.
> > >
> > > In practice, NPPR is often more useful when restricted to a smaller, application-informed set $\mathcal{Q}\subseteq\mathcal{P}_\epsilon$ that captures uncertainty about the operational environment. For example, for a roadside camera at night, test-time perturbations may come from mild sensor noise, packet-loss-like impulse corruption, or vibration/rain-induced blur. The assessor may know that the perturbation comes from this family, while not knowing which mechanism or mixture will dominate at test time. In this setting, the **distinct diagnostic value** of NPPR is that it asks: **what is the worst probabilistic robustness the model can guarantee over all plausible perturbation distributions in the deployment environment?** This ambiguity-set view is the main practical value we intend to emphasize, and we will clarify this more clearly in the revision.
> > >
> > > ### Comparison with weak-adversary and intermediate-stress baselines
> > >
> > > We compare NPPR on ResNet18 with a weak-adversary baseline (1-step PGD with random Gaussian initialization, variance $0.1$) and with the intermediate-stress formulation of Rice et al. [1], with the latter normalized by the standard loss for easier comparison. The results show that NPPR behaves differently from the other two baselines, suggesting that it is not simply another tuned intermediate-severity score.
> > >
> > > |Dataset|NPPR|var=0.1|$q=1$/std|$q=10$/std|$q=100$/std|$q=1000$/std|
> > > |---|---:|---:|---:|---:|---:|---:|
> > > |CIFAR10|0.76|0.87|0.18|0.54|0.86|0.91|
> > > |CIFAR100|0.59|0.72|0.36|0.62|0.92|0.98|
> > > |TinyImageNet|0.53|0.77|0.63|0.75|0.98|1.04|
> > >
> > > We will also clarify in the revision that Rice et al. (2021) studies robustness notions between average-case and worst-case evaluation under a given perturbation measure, whereas NPPR is motivated by uncertainty over the perturbation distribution itself and evaluates robustness conservatively over an admissible family of perturbation distributions.
> > >
> > > ## Regarding the perturbation family design and its distinction from attack-based evaluation
> > >
> > > We agree that this distinction was not sufficiently clear. The low-entropy behavior noted by the reviewer does not by itself imply that NPPR reduces to an amortized attack prior. Under the full set $\mathcal P_\epsilon$, such concentration is consistent with the conservative objective of NPPR. Under a practical set $\mathcal{Q}\subseteq\mathcal{P}_\epsilon$, it instead indicates that the model is particularly sensitive to one plausible perturbation mechanism within the operational family.
> > >
> > > Moreover, NPPR does **not inherently require** using the evaluated classifier’s own feature representation. The perturbation-family model can be learned in a different feature space from that of the target classifier, which makes the framework different from standard attacks optimized directly against one specific model. To illustrate this, we evaluate NPPR on CIFAR10 under $L_\infty(16/255)$ using different feature extractors for the GMM head. Rows denote the feature extractor used to train the estimator, and columns denote the target classifier being evaluated.
> > >
> > > |GMM feat.\Classifier|ResNet18|ResNet50|VGG16|WRN|
> > > |---|---:|---:|---:|---:|
> > > |ResNet18|76.27|80.12|74.99|77.26|
> > > |ResNet50|86.41|86.81|86.03|86.09|
> > > |VGG16|76.60|80.66|74.47|77.84|
> > > |WRN|82.40|84.21|79.79|81.32|
> > >
> > > In the revision, we will make these points explicit by (i) narrowing the claim of NPPR to a conservative probabilistic robustness quantity under perturbation-distribution uncertainty, (ii) clarifying its distinction from attack-based evaluation, and (iii) adding a more direct comparison to Rice et al. (2021) and the weak-adversary/intermediate-stress baselines.
> > >
> > > [1] Rice, Leslie, et al. "Robustness between the worst and average case." Advances in Neural Information Processing Systems 34 (2021): 27840-27851.

---

### Official Review · Reviewer_q1Jx · 2026-03-02

**Soundness:** 3
**Presentation:** 3
**Significance:** 2
**Originality:** 3
**Overall Recommendation:** 4
**Confidence:** 4

**Summary:**

This paper proposes a novel metric for evaluating the robustness of deep learning models: Nonparametric Probabilistic Robustness (NPPR). The authors developed an NPPR estimator using a Gaussian mixture model (GMM) combined with the Gumbel-Softmax technique and addressed the computational efficiency problem in high-dimensional image spaces through bicubic upsampling.

**Compliance With Llm Reviewing Policy:**

Affirmed.

**Final Justification:**

After reading the responses and other information, I will keep my original rate.

**Key Questions For Authors:**

See the weakness

**Limitations:**

The generalization and computational overhead should be discussed.

**Strengths And Weaknesses:**

Strength：

1. This paper  addresses the weakness of reliance on known distribution assumptions. The introduction of NPPR provides a more conservative and robust perspective for robustness assessment.

2. The experimental results are solid.

Weakness:

1. The generalization ability between different radii is unclear. Although Table 3 exists, I am curious to know how an NPPR estimator trained with one radius performs when evaluated with another radius.

2. Compared to standard PR, which only requires simple Monte Carlo sampling, NPPR requires training a complex GMM estimator. Although the authors mention using a lightweight MLP, the sensitivity of training overhead on large-scale datasets still needs to be discussed in more detail.

2. While NPPR can learn the worst-case random distribution, it is still fundamentally based on statistical expectation. Can NPPR remain sufficiently vigilant against certain spatially small but highly destructive "outlier" attacks? It is recommended that the discussion incorporate a link to certified robustness.

---

> ### Author Rebuttal · Authors · 2026-03-31
>
> We sincerely thank the reviewer for the positive assessment of our paper, especially for recognizing the motivation of relaxing the known-distribution assumption, the conservative perspective offered by NPPR, and the overall strength of the experimental results. We also thank the reviewer for the constructive comments on cross-radius generalization, computational overhead, and highly destructive outlier attacks.
>
> ### (W.1) On generalization across different perturbation radii.
> We sincerely thank the reviewer for raising this insightful question. We believe this is a very interesting point, as it helps clarify how the learned perturbation distribution interacts with the perturbation budget in our current estimator.
>
> In fact, cross-radius evaluation is **possible** under our framework, but the current projection mechanism makes the behavior rather constrained. Specifically, the GMM is learned under a fixed training radius $\epsilon_{\mathrm{train}}$, so the learned perturbation distribution tends to concentrate within that budget. As a result, when evaluated with a larger radius, the sampled perturbations still mostly remain concentrated around the scale learned at $\epsilon_{\mathrm{train}}$, and the resulting PR value therefore changes little. On the other hand, when evaluated with a smaller radius, the sampled perturbations are projected back into the smaller admissible set, which again tends to limit the change in the resulting PR estimate. In this sense, the projection mechanism in the current estimator largely determines the cross-radius behavior.
>
> We agree with the reviewer that this is an important and interesting issue. In the revised version, we will clarify this point explicitly in the main text and add a discussion in the appendix to explain how the current projection mechanism affects cross-radius generalization. We thank the reviewer again for highlighting this subtle but valuable aspect of the method.
>
> ### (W. 2) On computational overhead relative to standard PR
> We sincerely thank the reviewer for highlighting this point. We agree that computational overhead is an important issue that should be discussed more explicitly. Although the GMM head in our estimator is designed to be lightweight, the overall training process can still be time-consuming in practice, especially when the number of mixture components $K$ becomes larger.
>
> In the revised version, we will make this limitation explicit in the main text, and report the practical training time together with the key hyperparameter settings in the appendix. We will also discuss a possible way to address this issue more effectively: instead of reusing the target model's own features, one may employ a separate feature extractor for the perturbation estimator. Such a design would only need to be trained once and could then be reused across different target models, making the framework more scalable in practice. We will include a more detailed discussion of this direction in the revised version.
>
> ### (W. 3) On outlier attacks and the relation to certified robustness
> We sincerely thank the reviewer for this thoughtful and insightful comment. We fully agree that this distinction is important. By construction, NPPR is an expectation-based robustness metric: it quantifies the minimum probability of preserving correct prediction over an admissible family of stochastic perturbation distributions, providing statistical guarantees. As such, its purpose is different from worst-case or certified robustness, which aim to guarantee correctness against every perturbation in a prescribed set.
> For this reason, NPPR should not be viewed as a replacement for certified robustness, but rather as a complementary notion. NPPR is intended for settings where perturbations are stochastic and their exact distribution is unknown, whereas certified robustness is designed to provide deterministic worst-case guarantees. Therefore, highly destructive but very rare outlier perturbations are indeed better captured by worst-case or certified formulations. In the revised version, we will make this distinction more explicit and add a brief discussion clarifying that NPPR complements AR/certified robustness by filling the gap between fixed-distribution PR and fully worst-case guarantees. We will also add a brief discussion connecting NPPR to certified robustness. In particular, certified robustness provides deterministic worst-case guarantees over a prescribed perturbation set, which is conceptually different from the expectation-based guarantee of NPPR. We will clarify this and add a brief discussion of representative certified robustness methods, including convex-relaxation approaches and randomized smoothing [1, 2].
>
> [1] Eric Wong and J. Zico Kolter. Provable Defenses against Adversarial Examples via the Convex Outer Adversarial Polytope. In ICML, 2018.
>
> [2] Jeremy Cohen, Elan Rosenfeld, and Zico Kolter. Certified Adversarial Robustness via Randomized Smoothing. In ICML, 2019.

---

> > ### Author Rebuttal · Reviewer_q1Jx · 2026-04-01
> >
> > The authors admit the limitaion and my concern, however, without evidence, I cannot guarantee that the author will reasonably address these issues in the revised version. Thus, I keep my original score unchanged.

---

> > > ### Author Response · Authors · 2026-04-01
> > >
> > > We sincerely thank the reviewer again for the thoughtful follow-up. We understand the concern that our rebuttal mainly described intended revisions without direct empirical evidence. To address this more concretely, we conducted additional experiments after the rebuttal stage. These directly target the two main issues raised by the reviewer: *cross-radius behavior* and *training overhead* of the NPPR estimator. All experiments below are based on GMM with 7 modes and conditioned on both inputs and labels.
> > >
> > > ### (1) Cross-radius behavior
> > > We agree this is an important question. To examine it directly, we made a small modification to the upsampler so that the learned estimator can be evaluated more flexibly across different radii.
> > >
> > > **Table 1. NPPR (%) trained and evaluated under different $L_\infty$ radii on ResNet18/CIFAR10**
> > > All values are percentages, i.e., estimated NPPR scores multiplied by 100.
> > >
> > > |Train\Eval|4/255|8/255|16/255|32/255|
> > > |---|---:|---:|---:|---:|
> > > |4/255|96.56|91.58|79.63|59.79|
> > > |8/255|96.77|91.31|77.66|55.60|
> > > |16/255|96.75|90.99|76.27|52.49|
> > >
> > > Table 1 shows two consistent patterns. First, for a fixed training radius, NPPR decreases monotonically as the evaluation radius increases, which matches the expected behavior under a larger perturbation budget. Second, estimators trained at different radii remain reasonably stable across evaluation radii. In particular, training with a larger radius tends to give slightly more conservative estimates, especially at larger evaluation radii (e.g., 52.49 vs. 59.79 at 32/255).
> > > To test whether this is specific to one architecture/dataset pair, we also fixed the training radius at 16/255 and evaluated across multiple models and datasets.
> > >
> > > **Table 2. NPPR (%) trained at 16/255 and evaluated under different $L_\infty$ radii**
> > > All values are percentages.
> > >
> > > |Arch|Dataset|4/255|8/255|16/255|32/255|
> > > |---|---|---:|---:|---:|---:|
> > > |ResNet18|CIFAR10|96.75|90.99|76.27|52.49|
> > > |ResNet18|CIFAR100|95.71|84.10|58.60|27.74|
> > > |ResNet18|TinyImageNet|93.24|78.43|53.29|27.09|
> > > |ResNet50|CIFAR10|98.51|94.99|86.81|68.41|
> > > |ResNet50|CIFAR100|96.03|84.46|58.92|27.49|
> > > |ResNet50|TinyImageNet|95.37|85.10|64.37|31.75|
> > > |VGG16|CIFAR10|96.27|90.06|74.47|49.72|
> > > |VGG16|CIFAR100|94.26|79.92|52.20|23.18|
> > > |VGG16|TinyImageNet|94.94|82.64|59.00|25.51|
> > > |WRN50|CIFAR10|97.76|93.15|81.32|59.45|
> > > |WRN50|CIFAR100|94.06|82.19|58.83|28.26|
> > > |WRN50|TinyImageNet|94.83|83.40|59.86|28.89|
> > >
> > >
> > > Table 2 shows the same monotone trend across all tested architectures and datasets: larger evaluation radii consistently lead to lower NPPR estimates, even when the estimator is trained at a smaller radius. These results suggest that the learned NPPR estimator remains stable under train/test radius mismatch.
> > >
> > > ### (2) Training overhead
> > >
> > > We also agree that computational overhead should be discussed more explicitly, especially relative to standard PR, which only requires Monte Carlo sampling under a fixed perturbation distribution.
> > >
> > > To provide a practical reference, we measured the wall-clock training time of the GMM-based NPPR estimator on a single RTX 3090. Since this measurement was conducted using one training epoch, we report the **per-epoch wall-clock time** below.
> > >
> > > **Table 3. Per-epoch wall-clock training time (s) of the NPPR estimator on a single RTX 3090**
> > >
> > > |Dataset|Arch|Epoch time(s)|
> > > |---|---|---:|
> > > |CIFAR10|ResNet18|69.453|
> > > |CIFAR10|ResNet50|154.785|
> > > |CIFAR10|WRN|228.243|
> > > |CIFAR10|VGG16|207.230|
> > > |CIFAR100|ResNet18|64.627|
> > > |CIFAR100|ResNet50|154.412|
> > > |CIFAR100|WRN|229.047|
> > > |CIFAR100|VGG16|195.081|
> > > |TinyImageNet|ResNet18|203.238|
> > > |TinyImageNet|ResNet50|565.148|
> > > |TinyImageNet|WRN|907.584|
> > > |TinyImageNet|VGG16|994.211|
> > >
> > > These results confirm the reviewer’s concern that NPPR introduces nontrivial fitting cost, and that the cost grows with both dataset size/input dimensionality and backbone complexity. We will make this limitation explicit in the revised paper.
> > >
> > > At the same time, we also tested whether a smaller or shared feature extractor can reduce this cost while still giving comparable robustness estimates.
> > >
> > > **Table 4. NPPR (%) on CIFAR10 under $L_\infty(16/255)$, using different feature extractors for the GMM head**
> > > Rows: feature extractor used to train the estimator; columns: target classifier being evaluated.
> > >
> > > |GMM feat.\Classifier|ResNet18|ResNet50|VGG16|WRN|
> > > |---|---:|---:|---:|---:|
> > > |ResNet18|76.27|80.12|74.99|77.26|
> > > |ResNet50|86.41|86.81|86.03|86.09|
> > > |VGG16|76.60|80.66|74.47|77.84|
> > > |WRN|82.40|84.21|79.79|81.32|
> > >
> > > Table 4 suggests that the estimator does not necessarily need to rely on the largest target model as its feature backbone. Smaller feature extractors can still yield reasonably competitive NPPR estimates in many cases, suggesting a practical direction for reducing training overhead.
> > >
> > > Overall, these additional results suggest that: (i) NPPR transfers stably across radii, and (ii) its extra fitting cost over standard PR may be reduced with a lighter backbone.

---

### Official Review · Reviewer_YMuy · 2026-03-12

**Soundness:** 3
**Presentation:** 3
**Significance:** 2
**Originality:** 2
**Overall Recommendation:** 4
**Confidence:** 3

**Summary:**

This paper considers the limitation of probabilistic robustness, which refers to the fixed distribution of perturbation, and then proposes non-parametric probabilistic robustness (NPPR). NPPR learns an optimized perturbation distribution, which covers more perturbation scenarios when incorporated with Gaussian Mixture Model.

**Compliance With Llm Reviewing Policy:**

Affirmed.

**Final Justification:**

Thank authors for their rebuttal. The authors provide more experimental results, but the obvious limitations of this work remain. Thus, I will raise the score but will ultimately take the other reviewers' comments into account.

**Key Questions For Authors:**

1. Since the proposed method aims at fewer dependency on specific distributions, I am concerned about the defense generalizations across various attacks like AA.

2. Recently, the robustness evaluations tend to tasks different from classifications, which could be discussed in brief.

3. When the perturbation dynamically changes, could the proposed method adapt to this scenario using some strategies?

**Limitations:**

Yes.

**Strengths And Weaknesses:**

Strengths:

1. The viewpoint of probabilistic robustness provides more theoretical guarantees, and this is precisely what empirical defense lacks.

2. The provided theoretic relationship of Probabilistic Robustness and Adversarial Robustness is inspirational for defenses.

3. The incorporated NPPR is flexible across various scenarios with acceptable computational efficiency.

Weaknesses:
1. The proposed method seems to lack the comparison with advanced methods. Since the paper only introduces traditional AR with PGD and PR baselines, it is difficult to intuitively reflect the performance positioning of NPPR in current research. Perhaps, the author can provide more discussions about the superiority formally.

2. The time cost enlarges when conducting evaluations on larger models and datasets. To this end, the ability to generalize across model and datasets with different scales is somewhat weak.

3. The limitation could be discussed more explicitly.

4. Some minors need to be noticed: in the last paragraph of Section 2.1: “in contract” -> “in contrast”.

---

> ### Author Rebuttal · Authors · 2026-03-31
>
> We sincerely thank the reviewer for the positive assessment and constructive suggestions. Below, we jointly address the related concerns and questions.
>
> **(W1 & Q1) Positioning of NPPR and comparison with stronger evaluations.**
> We address Weakness 1 and Key Question 1 together, since both concern how NPPR should be positioned relative to stronger evaluation methods such as AA. NPPR is **not** intended as a defense method or a stronger attack, it is an **evaluation framework under unknown perturbation distributions**. AR methods such as PGD/CW/AA measure robustness against optimized deterministic perturbations, whereas NPPR estimates the **minimum probabilistic robustness** over an admissible set of stochastic perturbation distributions. These are complementary rather than competing notions.
>
> To make this positioning clearer, we additionally compared NPPR with fixed-distribution PR baselines, strong AR evaluations, and common corruptions. A consistent pattern is that NPPR tracks the relative corruption difficulty across model--dataset pairs, while fixed-distribution PR remains overly optimistic and AR measures a different worst-case notion. We will include the full results in the revised paper. A compact example (ResNet18) is shown below, all values are **normalized accuracies**, i.e., evaluated on the subset of cleanly and correctly classified samples.
>
> | Dataset | PR-Gauss | PR-Uniform | NPPR | PGD | CW | AA | Salt\&Pepper | MotionBlur |
> |---|---:|---:|---:|---:|---:|---:|---:|---:|
> | CIFAR-10 | 0.9897 | 0.9928 | 0.7629 | 0.0008 | 0.0013 | 0.0000 | 0.8363 | 0.7365 |
> | CIFAR-100 | 0.9712 | 0.9819 | 0.5862 | 0.0044 | 0.0060 | 0.0006 | 0.6903 | 0.6018 |
> | Tiny-ImageNet | 0.9825 | 0.9898 | 0.5321 | 0.0000 | 0.0000 | 0.0000 | 0.7161 | 0.3378 |
>
> These results support our intended claim: NPPR provides a practical and conservative robustness estimate when the perturbation distribution is unknown. We will revise the paper to make this positioning much more explicit.
>
> **(W2) Scalability and time cost.**
> We agree that the evaluation cost increases with model and dataset scale. This is expected because NPPR learns a conservative perturbation distribution rather than assuming a fixed one. However, the added overhead over standard PR is moderate, and NPPR remains much faster than strong attack suites such as AA. We will add the full timing results in the revision. A compact summary is shown below.
>
> | Arch / Dataset | PR-Gauss | NPPR | AA |
> |---|---:|---:|---:|
> | ResNet18 / CIFAR-10 | 2.55s | 3.96s | 30.20s |
> | ResNet50 / Tiny-ImageNet | 20.98s | 23.60s | 143.27s |
> | WRN50-2 / Tiny-ImageNet | 34.29s | 37.16s | 244.73s |
>
> We will clarify in the revision that NPPR introduces only a modest overhead over standard PR while providing a more practical evaluation setting under unknown perturbation distributions.
>
> **(W3 & Q2) Limitations and tasks beyond classification.**
> We address Weakness 3 and Key Question 2 together. We agree that the limitations should be stated more explicitly. In the revision, we will clarify that: (i) the current estimator is based on a finite GMM approximation and may not fully cover the entire admissible distribution set; and (ii) the current experiments focus on image classification.
>
> At the same time, NPPR is not inherently limited to classification. In principle, it can be extended to other tasks once a suitable task-specific robustness criterion is defined as indicated by [1], it can extend to **detection, segmentation, regression and text2image, etc..** We will discuss this in our revised paper.
>
> **(W4 & Q3) Minor issue and dynamically changing perturbations.**
> Thank you for catching the typo in Section 2.1; we will correct “in contract” to “in contrast”.
>
> Regarding dynamically changing perturbations, yes, NPPR is conceptually compatible with this scenario. Since NPPR is defined over an admissible perturbation distribution set, it remains conservative as long as the true perturbation distribution changes **within** that set. If the perturbation regime moves outside the admissible set, NPPR should be re-estimated using updated data. One potntial use case of NPPR is to cover this scenarios: we have a fairly broad set of all possible changes distributions, and when indeed the perturbation distributions changes in the admission set, then NPPR estimate is still gurarteend to be conservative; otherwise, NPPR needs to be recalculated based on new data. We will add this point briefly in the revision as an important future direction.
>
>
> [1] Zhao, Xingyu. "Probabilistic robustness in deep learning: A concise yet comprehensive guide." Adversarial Example Detection and Mitigation Using Machine Learning. Cham: Springer Nature Switzerland, 2026. 209-222.

---

> > ### Author Rebuttal · Reviewer_YMuy · 2026-04-02
> >
> > Thank authors for their rebuttal. The authors provide more experimental results, but the obvious limitations of this work remain. Thus, I will raise the score but will ultimately take the other reviewers' comments into account.

---

> > > ### Author Response · Authors · 2026-04-08
> > >
> > > Thank you for the update and for reconsidering the score. We appreciate that the additional experiments helped clarify our intended positioning. We will further refine the presentation in the revision to make this positioning as clear as possible.

---

### Official Review · Reviewer_4BRS · 2026-03-13

**Soundness:** 2
**Presentation:** 3
**Significance:** 3
**Originality:** 2
**Overall Recommendation:** 4
**Confidence:** 3

**Summary:**

This paper studies robustness evaluation and proposes NPPR, a robustness metric that does not assume a known perturbation distribution. Instead of evaluating probabilistic robustness (PR) under a fixed distribution (e.g., Gaussian), NPPR searches for the worst-case distribution within an admissible set via a learned GMM with MLP heads. The authors present a hierarchy of robustness metrics (Prop.3.4), provide SGD convergence guarantees (Thm.4.2), and validate on CIFAR-10/100 and Tiny ImageNet across four architectures.

**Compliance With Llm Reviewing Policy:**

Affirmed.

**Final Justification:**

After reading the answers of the authors, i upgraded my score

**Key Questions For Authors:**

Question 1: Can you quantify theroretically and/or experimentally how the parametric GMM class approximates the non-parametric set of all distributions and how does this affect the conservativeness of NPPR?

Question 2: Can you clarify the definition of AR, PR, and NPPR so that they have consistent interpretation?

**Limitations:**

yes

**Strengths And Weaknesses:**

The assumption of a known perturbation distribution in PR is unrealistic. The paper clearly articulates why this matters, and therefore the proposed solution is well-motivated. The empirical study spans several datasets, architectures and perturbation settings, which suggests that the method is broadly applicable.

However, the non-parametric framing is misleading since the method optimizes over a GMM with fixed K components. Theoretical results or discussion on the expressivity of this parametric class are needed, as this seems to be the core of the paper. Instead, the actual theoretical results are quite standard (Gumbel-Softmax approximation, SGD convergence) and do not give insight into the expressivity of this class. Also, the definitions of AR, PR, and NPPR are not consistent, as high AR leads to low robustness while high PR leads to high robustness, which causes a soundness issue in the core theoretical result (Prop.3.4).

---

> ### Author Rebuttal · Authors · 2026-03-31
>
> We sincerely thank the reviewer for the careful reading and constructive comments. We address the two questions below.
>
> **Q1: How does the parametric GMM class approximate the non-parametric distribution set, and how does this affect conservativeness?**
> We agree that the current submission does not clearly distinguish the **ideal NPPR objective** from our **current GMM-based estimator**. Our intended meaning of “non-parametric” is at the **metric level**: NPPR is defined as the infimum of PR over the full admissible perturbation distribution set, without assuming a fixed perturbation family a priori. The current implementation, however, uses a finite-$K$ GMM as a practical estimator of this ideal objective and the choice of $K$ component is of practical interest, as it may represent $K$ most common noise for the given application (e.g., fog, motion blur, dark-light etc.), as in real-life, perturbations could be a mixture of $K$ common types of noises. In the revision, we will clarify this point explicitly and describe our method as a **parametric approximation to the ideal NPPR objective**.
> Formally, let
> $$
> \mathfrak S_{\mathrm{NPPR}}(x,y) := \inf_{\omega\in\mathcal P_\varepsilon} \mathbb E_{\varepsilon\sim\omega(\cdot\mid x, y)} \big[\mathbf 1_{h(x+U(\varepsilon))=y}\big]
> $$
> be the ideal NPPR value over admissible perturbation distribution set, and let
> $$
> \mathfrak S_{K}(x,y) := \inf_{\omega\in\mathcal Q_{K}} \mathbb E_{\varepsilon\sim\omega(\cdot\mid x, y)} \big[\mathbf 1_{h(x+U(\varepsilon))=y}\big]
> $$
> be the value restricted to the $K$-component GMM family $\mathcal Q_K\subseteq\mathcal P_\varepsilon$. Then
> $$
> \mathfrak S_{\mathrm{NPPR}}(x,y)\leq \mathfrak S_K(x,y)\leq\mathfrak S_{\mathrm{PR}}(x,y).
> $$
> where $\mathfrak S_{\mathrm{PR}}(x,y)$ is estimated by standard Gaussian. Thus, the finite-GMM estimator remains more conservative than standard PR, but can be less conservative than the fully unrestricted NPPR optimum due to the family restriction. To quantify this gap, define
> $$
> \eta_K := \sup_{\omega\in\mathcal P_\varepsilon} \inf_{\widetilde\omega\in\mathcal Q_K} d_{\mathrm{TV}}(\omega,\widetilde\omega).
> $$
> Since the indicator loss is bounded in $[0,1]$, we have
> $$
> 0 \leq \mathfrak S_K(x,y)-\mathfrak S_{\mathrm{NPPR}}(x,y)\leq \eta_K,
> $$
> and therefore
> $$
> 0 \leq \mathcal G_K-\mathcal G_{\mathrm{NPPR}}\leq \eta_K.
> $$
> Hence, the conservativeness gap is controlled by how well the finite-$K$ GMM family approximates the worst admissible perturbation distribution.
>
> Empirically, this trend is consistent with our results: in Table 2, increasing the number of mixture modes generally leads to smaller robustness values. Under joint dependence, the value decreases from $90.04$ at $K=3$ to $89.72$ at $K=7$, and to $89.50$ at $K=12$. We also performed an additional training-stage analysis on a small CIFAR-10 subset (ResNet18, max batch $=20$), where PR further decreases with longer optimization and richer mixtures: from $0.4960$ to $0.3357$ for $K=1$, and from $0.3892$ to $0.2530$ for $K=3$ (In Tab. 1, below). This supports the interpretation above. In the revision, we will add this clarification and a short proposition formalizing the approximation gap.
>
> **Table 1.**
> |$K$|Epoch|Loss|PR|
> |-----|-------|--------|--------|
> |1|100|2.03|0.49|
> |1|300|1.60|0.38|
> |1|500|1.42|0.33|
> |3|100|1.60|0.38|
> |3|300|1.26|0.30|
> |3|500|1.08|0.25|
>
> **Q2: Can AR, PR, and NPPR be defined with a consistent interpretation?**
> We agree. The current presentation is not fully consistent because AR is written as a worst-case loss, while PR and NPPR are written as robustness scores. This is mainly a notation issue, and we will fix it explicitly.
>
> Our intention throughout the theory and experiments was to use AR in the same monotone direction as PR and NPPR, namely as a robustness score. We will revise the local AR definition to
> $$
> \mathfrak S_{\mathrm{AR}}(x,y) := 1-\max_{\varepsilon\in\mathcal B}\mathbf 1_{h(x+\varepsilon)\neq y} \in \{0,1\},
> $$
> where $\mathcal B$ is the perturbation budget. Under this definition, $\mathfrak S_{\mathrm{AR}}(x,y)=1$ if the classifier is correct for all admissible perturbations, and $0$ otherwise. Hence, $\mathfrak S_{\mathrm{AR}}$, $\mathfrak S_{\mathrm{PR}}$, and $\mathfrak S_{\mathrm{NPPR}}$ all have the same interpretation: larger values indicate higher robustness.
>
> Under this unified convention, Proposition 3.4 becomes
> $$
> \mathcal G_{\mathrm{AR}} \leq \mathcal G_{\mathrm{NPPR}} \leq \mathcal G_{\mathrm{PR}},
> $$
> and if $\mathcal P_\varepsilon$ is unrestricted and includes Dirac measures, then
> $$
> \mathcal G_{\mathrm{AR}}=\mathcal G_{\mathrm{NPPR}}.
> $$
> Thus, the core ordering result is unchanged. The revision only makes the notation and interpretation consistent throughout the paper. We will revise Definition 2.1, Figure 2, Proposition 3.4, and the related discussion accordingly. We believe that this is a minor notation issue that can be easily fixed without any negative impact on the main contributions.

---

> > ### Author Rebuttal · Reviewer_4BRS · 2026-04-02
> >
> > Thank you for you answer. Also, you write in the paper that "All aforementioned state-of-the-arts rely on a shared assump-
> > tion that the perturbation noise follows a predefined distribution, which is rarely known in practice". There are in fact papers that you do not cite which create a probabilistic attack dependent on the point being attacked, e.g. [Mixed Nash Equilibria in the Adversarial Examples Game].

---

> > > ### Author Response · Authors · 2026-04-07
> > >
> > > ## Response to the comment on related randomized / game-theoretic adversarial formulations
> > >
> > > Thank you for this valuable comment. We thank the reviewer for pointing out this relevant line of work. We agree that our original wording was too broad, especially the statement suggesting that prior works generally rely on a predefined perturbation distribution. This comment is very helpful for improving the positioning of our paper, and we will revise the manuscript accordingly.
> > >
> > > Regarding the paper *Mixed Nash Equilibria in the Adversarial Examples Game* [1], it studies adversarial robustness from a game-theoretic perspective, where both the attacker and the classifier may adopt randomized (mixed) strategies. Its focus is on analyzing adversarial risk through a zero-sum game formulation and characterizing mixed Nash equilibria. In this sense, it is clearly relevant to our work, since it goes beyond purely deterministic adversarial perturbations and considers distributional/randomized adversarial behavior.
> > >
> > > More broadly, following the reviewer’s suggestion, we also revisited related literature on randomized or game-theoretic adversarial robustness, and identified several additional relevant references, including *Randomization Matters: How to Defend Against Strong Adversarial Attacks* [2], *A Game Theoretic Analysis of Additive Adversarial Attacks and Defenses* [3], and *On the Role of Randomization in Adversarially Robust Classification* [4]. Together, these works further confirm that randomized adversaries/defenders and game-theoretic robustness formulations form an important related research direction. Although their primary focus differs from our NPPR metric, they are highly relevant for better contextualizing our contribution relative to prior studies that move beyond purely deterministic attack models.
> > >
> > > At the same time, there are important differences between this line of work and ours. In particular, *Mixed Nash Equilibria in the Adversarial Examples Game* [1] focuses on **adversarial risk minimization in a game-theoretic setting**, where randomization is part of the attacker/defender strategy space. In contrast, our paper focuses on **probabilistic robustness evaluation under unknown perturbation distributions**. More specifically, NPPR is defined as the infimum of probabilistic robustness over an admissible set of perturbation distributions, with the goal of providing a conservative robustness metric when the true perturbation distribution is not known a priori. Therefore, while the cited work and the above related papers are closely connected to our motivation and should indeed be discussed, they study related but distinct objectives and formulations from those of NPPR.
> > >
> > > In the revised version, we will make the following changes:
> > > 1. Include a discussion of *Mixed Nash Equilibria in the Adversarial Examples Game* and the above related works in the related work and preliminary sections.
> > > 2. Revise our wording to avoid an overly broad claim.
> > > 3. Clarify that our intended statement pertains to **most existing probabilistic robustness formulations**, rather than to all robustness-related works more broadly.
> > >
> > > Overall, we thank the reviewer again for this helpful suggestion. We believe that incorporating this line of work will strengthen the paper by positioning NPPR more accurately with respect to prior studies on randomized and distributional adversarial formulations. In particular, we will revise the sentence in Sec. 2.2 that currently states that prior works rely on a predefined perturbation distribution, and replace it with a more precise and appropriately qualified statement.
> > >
> > > [1] Meunier, Laurent, et al. "Mixed Nash Equilibria in the Adversarial Examples Game." *Proceedings of the 38th International Conference on Machine Learning (ICML)*. PMLR, 2021.
> > >
> > > [2] Pinot, Rafael, et al. "Randomization Matters: How to Defend Against Strong Adversarial Attacks." *Proceedings of the 37th International Conference on Machine Learning (ICML)*. PMLR, 2020.
> > >
> > > [3] Pal, Ambar, and René Vidal. "A Game Theoretic Analysis of Additive Adversarial Attacks and Defenses." *Advances in Neural Information Processing Systems* 33 (2020): 1345–1355.
> > >
> > > [4] Gnecco Heredia, Lucas, et al. "On the Role of Randomization in Adversarially Robust Classification." *Advances in Neural Information Processing Systems* 36 (2023): 79293–79319.

---

### Decision · Program_Chairs · 2026-04-30

**Decision:**

Accept (regular)

**Comment:**

The paper introduces a new way (NPPR: non-parametric probabilistic robustness) to estimate robustness of ML models, which falls between the worst case adversarial robustness and average case probabilistic robustness. Specifically, the paper removes the assumption of probabilistic robustness in which the perturbation distribution is fixed, and known, to us. In particular, they allow a family of perturbations, and then try to learn the worst one in that family using tools such as Gaussian Mixture Models (GMMS).

On the positive side, the reviewers appreciate a new approach to robustness that removes the "known distribution" of probabilistic robustness, and computational aspects and experiments are insightful along theoretical results that put NPPR between PR and Adversarial robustness.

Some limitations were raised by the reviewers (e.g., (1) how the paper distinct itself from previous approaches that connect adversarial robustness to probabilistic robustness, and (2) how the approach is actually non-parametric, as it seems to use parameters in learning the worst distribution in the family) but the rebuttal discussions led to clarifications (and scores were raised).